# AAV-mediated base-editing therapy ameliorates the disease phenotypes in a mouse model of retinitis pigmentosa

Yidong Wu[1,2,3,7], Xiaoling Wan [1,2,3,7] ✉, Dongdong Zhao [4,5,7], Xuxu Chen[4,5,7], Yujie Wang[4,5,7], Xinxin Tang[4,5], Ju Li [6], Siwei Li [4,5], Xiaodong Sun [1,2,3] ✉, Changhao Bi [4,5] ✉ & Xueli Zhang [4,5] ✉

Base editing technology is an ideal solution for treating pathogenic single-nucleotide variations (SNVs). No gene editing therapy has yet been approved for eye diseases, such as retinitis pigmentosa (RP). Here, we show, in the *rd10* mouse model, which carries an SNV identified as an RP-causing mutation in human patients, that subretinal delivery of an optimized dual adeno-associated virus system containing the adenine base editor corrects the pathogenic SNV in the neuroretina with up to 49% efficiency. Light microscopy showed that a thick and robust outer nuclear layer (photoreceptors) was preserved in the treated area compared with the thin, degenerated outer nuclear layer without treatment. Substantial electroretinogram signals were detected in treated *rd10* eyes, whereas control treated eyes showed minimal signals. The water maze experiment showed that the treatment substantially improved vision-guided behavior. Together, we construct and validate a translational therapeutic solution for the treatment of RP in humans. Our findings might accelerate the development of base-editing based gene therapies.

Single-nucleotide variations (SNVs) in the human genome may generate changes in protein sequences or change the properties of the original DNA and cause genetic diseases[1]. The diseases caused by SNVs include rare diseases, such as sickle cell disease, thalassemia, and retinitis pigmentosa (RP). A study by the European Organization of Rare Diseases has shown that more than 450 million people worldwide suffer from rare diseases, and 95% of these diseases are without effective treatments to date[2]. A newer-generation CRISPR technique, base editing (BE), enables the direct, irreversible correction of base mutations and has a promising future for curing SNV-caused genetic diseases[3,4]. Compared with standard genome editing, BE can effectively repair base mutations without double-stranded DNA breaks (DSBs), which reduces the occurrence of insertions or deletions (indels) at target sites.

RP, the most common form of inherited retinal diseases (IRDs), is a major cause of blindness in developed countries[5,6]. Patients typically experience night blindness at the early stage as rod photoreceptors are primarily affected. As rod degeneration progresses, the peripheral vision diminishes, resulting in a constricted field of view. Over time, the macular cones are also affected, causing a loss of daylight vision. Many patients with RP become legally blind by middle age.

The rod phosphodiesterase (PDE6) plays an essential role in the phototransduction cascade[7]. Upon light stimulation, PDE6 hydrolyzes intracellular cyclic guanosine monophosphate (cGMP), which leads to

[1]Department of Ophthalmology, Shanghai General Hospital, Shanghai Jiao Tong University School of Medicine, Shanghai, China. [2]National Clinical Research Center for Eye Diseases, Shanghai, China. [3]Shanghai Key Laboratory of Ocular Fundus Diseases, Shanghai, China. [4]Tianjin Institute of Industrial Biotechnology, Chinese Academy of Sciences, Tianjin, China. [5]National Technology Innovation Center of Synthetic Biology, Tianjin, China. [6]College of Life Science, Tianjin Normal University, Tianjin, China. [7]These authors contributed equally: Yidong Wu, Xiaoling Wan, Dongdong Zhao, Xuxu Chen, Yujie Wang. ✉e-mail: shaolin.72@163.com; xdsun@sjtu.edu.cn; bi_ch@tib.cas.cn; zhang_xl@tib.cas.cn

the closure of cyclic nucleotide-gated channels and thereby produces an electrical response[7]. Mutations in the gene encoding the β subunit of PDE6 (PDE6B) are a common cause of autosomal recessive RP, which is estimated to be responsible for 36,000 new cases worldwide each year[5,6,8,9]. The *rd10* mouse, which carries a missense mutation (c.1678C>T, p.R560C) in the *Pde6b* gene, resembling the phenotype of typical human RP patients[10,11], is one of the most popular models for advanced therapeutic testing[12–20]. Notably, the homologous point mutation has recently been identified as an RP-causing mutation in a Spanish family[21], further enhancing the translational value of the *rd10* model.

Several approaches have been developed to rescue the death of photoreceptor cells in the *rd10* model, among which genetic therapy may be the most promising. One straightforward strategy is to deliver wild-type (WT) copies of *Pde6b* cDNA into retinal cells to produce functional proteins, an approach termed gene augmentation. Previous studies reported variable efficacy ranging from partial or transient rescue[16–18] to more stable therapeutic effects[19]. This variability might be explained by multiple factors, such as the gene transcription promoter and the timing of treatment. An important consideration, however, is the level of transgene expression. Due to the enormous signal amplification and multifaceted regulation within the phototransduction cascade[7], artificial gene augmentation is unlikely to achieve optimal equilibrium protein levels. Furthermore, the introduced DNA expression cassettes may degrade over time, which causes the fading effect[22–24]. These issues can be solved by direct gene editing, which corrects the pathogenic mutation in situ, allowing physiological level expression, as the corrected gene is expressed within the genomic context and is subject to endogenous regulatory mechanisms[25]. This concept was first tested by Vagni et al. using a CRISPR/Cas9-induced homology-directed repair (HDR) strategy to correct the SNV in *rd10* mice through electroporation delivery[20]. However, the active Cas9 method might not be a suitable approach for clinical application, because the safety of Cas9 treatment is questionable. Cas9 generates DNA DSBs and induces high rates of undesired indel mutations[26] as well as immunotoxicity[27]. Additionally, correction by the HDR pathway relies on cell division and has been shown to be highly ineffective in nondividing cells, such as the photoreceptors[26].

In contrast to the standard CRISPR/Cas9 method, BE mediates targeted single-nucleotide conversions without involving DSBs and the HDR pathway[3,4]. Lentiviral vector-delivered base editors have been used to correct *Rpe65* mutation in a mouse model of Leber congenital amaurosis (LCA) type 2 (*rd12*) and successfully restored visual function[28]. However, due to the safety issue of the lentivirus system, these strategies might be challenging when applied to human patients.

In this work, we planned to utilize the BE technique to correct the pathogenic mutation of the *rd10* mouse, which is theoretically safer than active Cas9 methods, and to use AAV for delivery, which has been employed in many gene augmentation clinical trials, showing a favorable safety profile. We expect this work to provide valuable information to medical and research societies and accelerate the development of BE-based gene therapies.

## Results

### Design and optimization of the BE system for *rd10* SNV correction

The *rd10* mouse model carries c.1678C>T SNV on exon 13 of the *Pde6b* gene, causing a change from arginine to cystine (R560C). Based on the SNV base, a PAM-less genome editing adenine base editor (ABE), SpRY-ABE8e, was selected and used to correct the pathogenic SNV. Since the ideal DNA package length of an AAV is under 4.7 kb, we split the base-editor system into amino-terminal and carboxy-terminal halves and packed them into two AAV capsids for delivery, designated the dual-AAV system. Utilizing the intein mechanism, after the two DNA templates enter the nucleus, the two halves of the SpRY-ABE8e complex

protein are produced in the cytoplasm, which splices in trans and reconstitutes a whole SpRY-ABE8e complex.

To select the best editing system, we first constructed an HEK293T cell line carrying a DNA cassette on the genome that consisted of the pathogenic SNV *rd10* along with its context sequence, which served as the BE model. The SpRY-ABE8e dual-AAV plasmids with 7 sgRNAs targeting the *rd10* pathogenic SNV were transferred into the BE model, and the BE outcome was analyzed by deep sequencing (Fig. 1a, b). The sequencing analysis revealed that among the 7 tested sgRNAs, sgRNA7 showed the highest target editing efficiency and the lowest bystander editing (Fig. 1c and Supplementary Table 1). Due to the superior correction rate and relatively low bystander editing, we selected sgRNA7 with the SpRY-ABE8e dual-AAV to perform the BE therapy experiment in the *rd10* mouse model.

### Subretinal delivery of AAV carrying SpRY-ABE8e corrected pathogenic SNV

We selected the AAV5 serotype for implementing the editors, since it has been proven efficient when targeting photoreceptor[29,30], and used in an ongoing clinical trial of PDE6B augmentation (NCT03328130). We prepared two dual-AAV systems, one encoding SpRY-ABE8e with sgRNA7 as the testing group and one with a nontargeting sgRNA to serve as the control group, referred to as AAV-SpRY-ABE8e-A7 and AAV-SpRY-ABE8e-NT, respectively. We injected postnatal (P) day 14 *rd10* mice subretinally with either AAV-SpRY-ABE8e-A7, AAV-SpRY-ABE8e-NT containing 1 compound of AAV-N and AAV-C at a viral titer of $10^{12}$, or balanced salt solution (BSS). Three weeks after injection, we examined the neuroretina from the eyes of injected *rd10* mice to determine the editing efficiency (Fig. 2a).

Genomic DNA analysis from AAV-SpRY-ABE8e-A7-treated eyes showed up to 17.49% of only A-to-G conversion at the target adenine, with an average correction efficiency of $13.06 \pm 2.22\%$ (Fig. 2b,c, $n = 7$). The average efficiencies of editing of the target adenine with bystander editing, bystander editing only, and indels were $3.05 \pm 0.91\%$, $1.42 \pm 0.11\%$, and $0.51 \pm 0.18\%$ respectively (Supplementary Table 2). The AAV-SpRY-ABE8e-NT-treated group ($n = 6$) and BSS-treated group ($n = 6$) demonstrated undetectable editing efficiencies.

To determine the editing efficiency of PDE6B-expressing cells, we examined the concentration of *Pde6b* cDNA-corrected sequence. The target-adenine-only editing efficiencies of cDNA were up to 49.11%, with an average of $34.07 \pm 7.12\%$ (Fig. 2b, c, $n = 7$). The average efficiencies of target with bystander editing, bystander editing only, and indels were $9.61 \pm 3.08\%$, $1.27 \pm 0.31\%$, and $0.72 \pm 0.18\%$ respectively (Supplementary Table 2). The AAV-SpRY-ABE8e-NT and BSS groups had no detectable cDNA editing efficiencies, consistent with the genomic DNA results. To determine the off-target effects, the editing efficiencies at 8 potential off-target sites predicted with Cas-OFFinder[31] were examined. We did not detect off-target editing above the background level of the AAV-SpRY-ABE8e-NT-treated eyes and BSS-treated eyes, as shown in Fig. 2d and Supplementary Table 3 ($p > 0.05$, one-way ANOVA).

Taken together, these findings suggested that our dual-AAV system corrected the pathogenic *Pde6b* SNV with substantial efficiency in vivo, with low bystander and indel editing, and without significant off-target effects.

### Dual-AAV SpRY-ABE8e treatment restored PDE6B protein expression in *rd10* mice

To explore whether the correction of *Pde6b* SNV has translated into the restoration of PDE6B protein, we performed Western blot analysis. The neuroretina lysates from AAV-SpRY-ABE8e-A7-treated eyes showed the PDE6B band (Fig. 3a). In contrast, no detectable PDE6B was found in either AAV-SpRY-ABE8e-NT-treated or BSS-treated eyes. The correct localization of the rescued PDE6B in the outer segment of the photoreceptor was confirmed by immunohistochemistry (Fig. 3b). Notably,

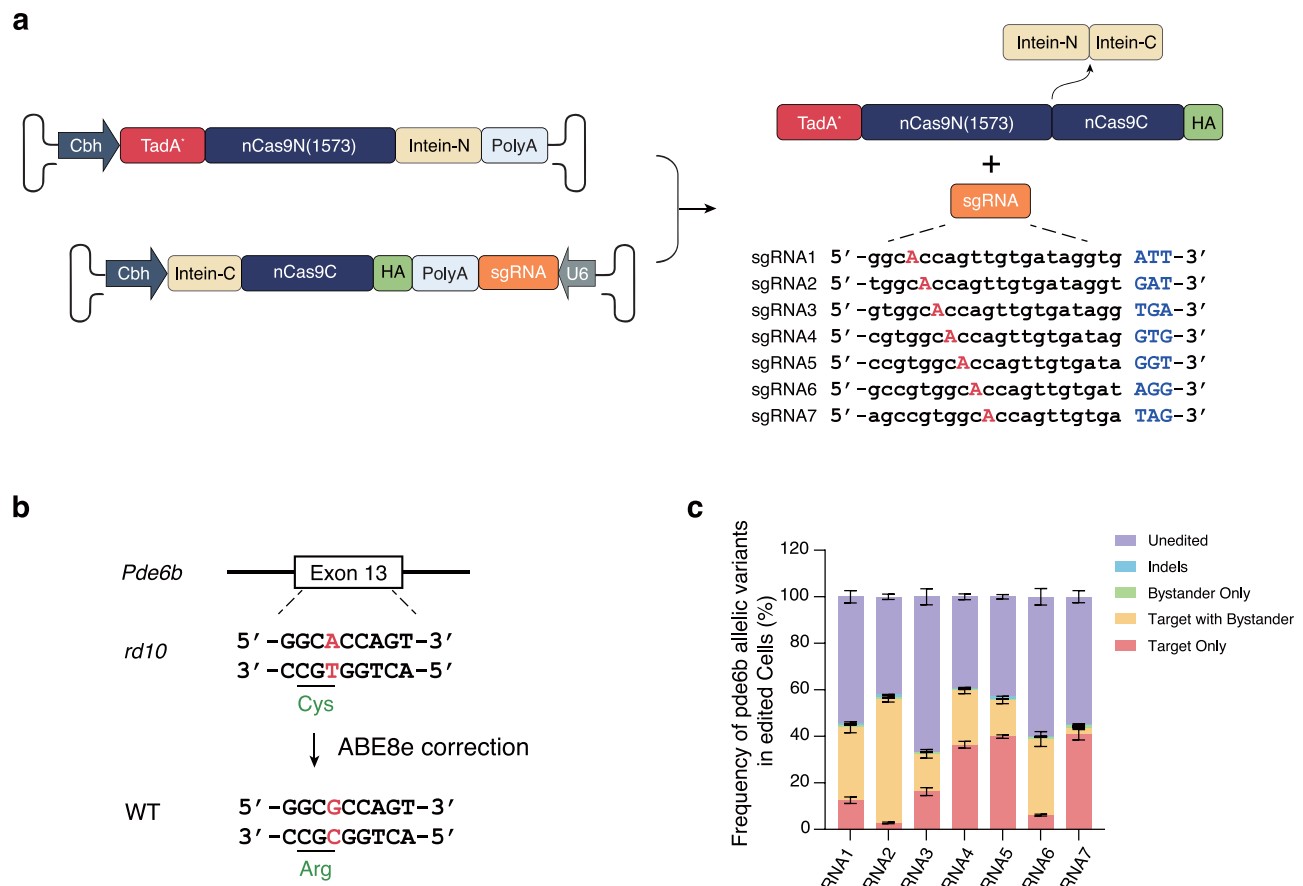

**Fig. 1 | Design and optimization of the ABE8 system for *rd10* SNV correction.** **a** Dual-AAV system of SpRY-ABE8e using a split-intein mechanism and sgRNA design targeting the *rd10* pathogenic SNV. **b** Schematic representation of *rd10* pathogenic SNV correction by SpRY-ABE8e. **c** BE efficiency of *rd10* pathogenic SNV by dual-AAV plasmids of SpRY-ABE8e. Bars represent the average editing specificity, and error bars represent the s.d. of three independent biological replicates. Source data are provided as a Source Data file.

both AAV-SpRY-ABE8e-A7- and AAV-SpRY-ABE8e-NT-treated eyes expressed SpRY-ABE8e protein in the photoreceptor nuclei, as shown in Fig. 3. These results indicated that dual-AAV SpRY-ABE8e treatment restores PDE6B protein expression in *rd10* mice.

### Dual-AAV SpRY-ABE8e treatment preserved the photoreceptors in *rd10* mice

A representative hematoxylin and eosin (H&E) staining image clearly revealed that the outer nuclear layer (ONL), i.e., photoreceptor nuclei, was largely preserved (6–7 rows) in the AAV-SpRY-ABE8e-A7-treated area, whereas only 1–2 rows remained in the untreated area of the same eye (Fig. 4a). The quantitative measurement on the retinal sections (Fig. 4b) indicated that the ONL was up to $37.62 \pm 3.06$ μm at the thickest position in the AAV-SpRY-ABE8e-A7-treated eyes ($n = 4$). This thickness was ~66% of the WT ($56.89 \pm 7.89$ μm, $n = 4$), in contrast to the BSS-treated ($n = 4$) and AAV-SpRY-ABE8e-NT-treated eyes ($n = 4$) where the ONL thickness dropped to averaging <11 μm (Fig. 4b).

To measure the rod outer segments (OS), we immunolabeled retinal sections with rhodopsin antibody (Fig. 4c). For AAV-SpRY-ABE8e-A7-treated eyes, the OS was clearly visible and the length ($7.25 \pm 0.7$ μm, $n = 4$) was ~50% of the WT ($14.53 \pm 1.5$ μm, $n = 4$) at thickest ONL portion (1 mm temporal of the optic nerve head; Fig. 4b, d). In BSS-treated and AAV-SpRY-ABE8e-NT-treated eyes, we could not identify typical OS structure, and we found that rhodopsin redistributed from outer segments to inner segments and photoreceptor cell bodies (Fig. 4c).

In *rd10* mouse, rod degeneration leads to secondary degeneration of cones. To further assess cone response to the treatment, we immunolabeled retinal sections with anti-cone arrestin antibody (Supplementary Fig. 1a). As expected, further cone degeneration was prevented in AAV-SpRY-ABE8e-A7-treated eyes while continued in BSS-treated and AAV-SpRY-ABE8e-NT-treated eyes.

Overall, in addition to the correction of the *Pde6b* gene and restoration of the PDE6B protein, our dual-AAV SpRY-ABE8e treatment also led to the preservation of the photoreceptor morphology.

### Dual-AAV SpRY-ABE8e treatment rescued retinal function in *rd10* mice

Next, we performed full-field electroretinography (ERG) to determine whether the restoration of PDE6B protein and preservation of photoreceptors results in the recovery of retinal function (Fig. 5a). To analyze the rod-dominated retinal function in more detail, we recorded scotopic ERGs in dark-adapted mice using a series of light stimuli increasing from −2 to 1.0 log cd s m$^{-2}$. The representative scotopic ERG traces and corresponding quantifications of the a- and b-wave amplitudes were shown in Fig. 5. The rod-dominated responses from age-matched WT mice showed an expected steady increase in amplitudes of both a- and b-waves, with the increasing stimulus intensity. As for the control *rd10* mice treated with BSS and AAV-SpRY-ABE8e-NT, the a-wave amplitudes attenuated to a negligible level, and b-wave amplitudes were also strongly reduced throughout the range of light stimuli. In the AAV-SpRY-ABE8e-A7-treated *rd10* mice, however, the b-waves were clearly visible at all stimulus intensities, and the a-waves were evident when the stimulus intensity rose up to 0 log cd s m$^{-2}$. The averaging a- and b-wave amplitudes were $66.11 \pm 22.27$ and $241.17 \pm 48.60$ μV for AAV-SpRY-ABE8e-A7-treated eyes ($n = 12$) as the

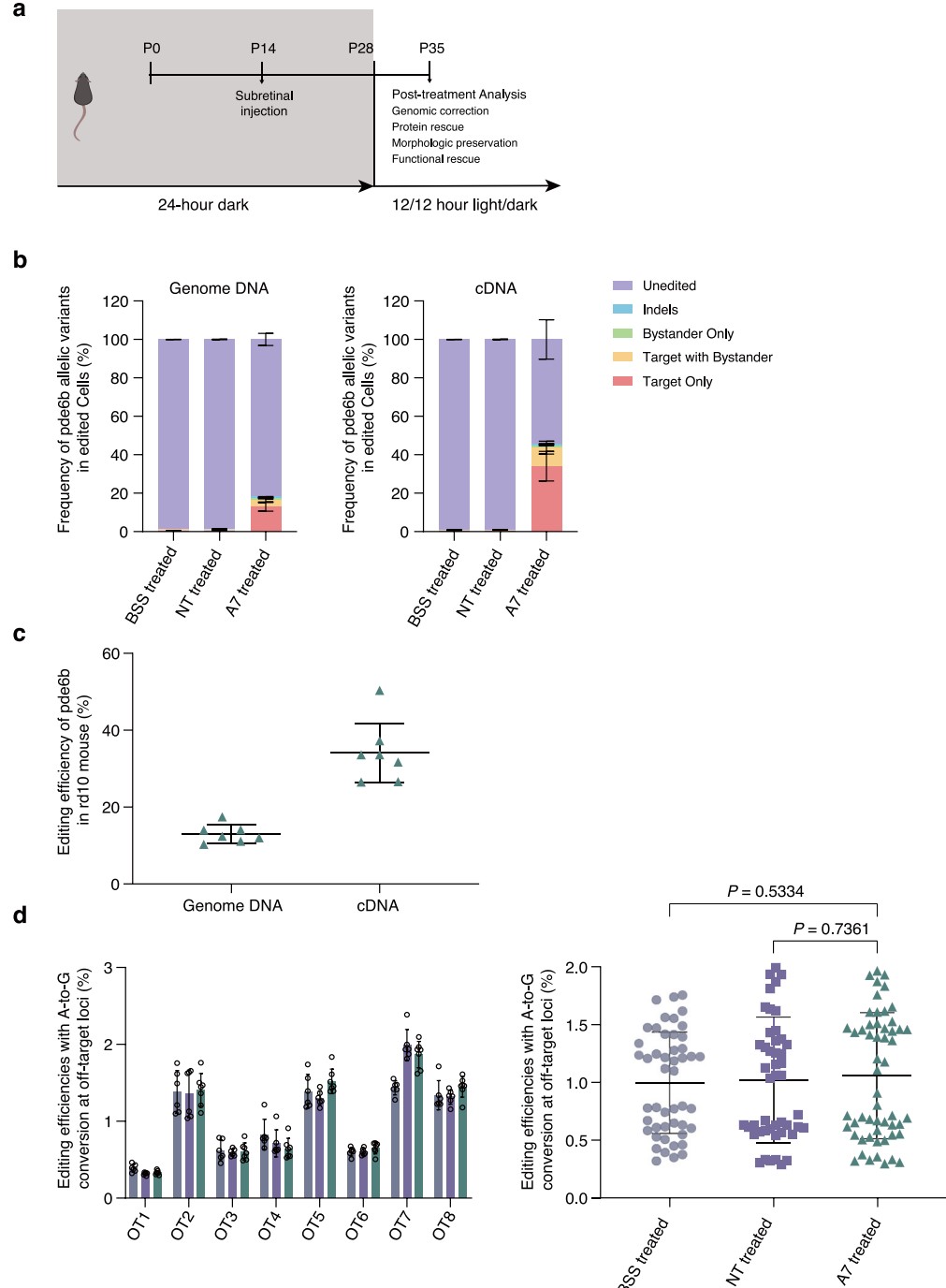

**Fig. 2 | Dual-AAV-mediated BE therapy in *rd10* mice. a** Flowchart of in vivo ABE treatment. **b** In vivo genomic DNA and cDNA BE efficiencies in *rd10* mice. **c** Comparison of average editing efficiencies of genomic DNA and cDNA. **d** Endogenous genome off-target efficiencies after SpRY-ABE8e treatment. Two-tailed unpaired *t*-tests; adjustment was not made for multiple comparisons. For all plots, the dots represent individual biological replicates, bars represent the average editing specificity, and error bars represent the s.d. of independent biological replicates. BSS treated (*n* = 6), NT treated (*n* = 6), and A7 treated (*n* = 7). Source data are provided as a Source Data file.

light intensity achieved 1.0 log cd s m$^{-2}$, which were ~21% and 47% of those from age-matched WT eyes (a-wave, 308.30 ± 55.57 μV; b-wave, 515.50 ± 125.14 μV; *n* = 10).

Since we have shown that the AAV-SpRY-ABE8e-A7 treatment preserved cone structure as well as rods (Supplementary Fig. 1a), we further recorded photopic ERG at 0.5 log cd s m$^{-2}$ to assess the cone function. It was again obvious that AAV-SpRY-ABE8e-A7-treated *rd10* mice showed significantly higher a- and b-wave amplitudes than BSS or AAV-SpRY-ABE8e-NT-treated *rd10* mice (Supplementary Fig. 1b,c). The averaging amplitudes (a-wave, 20.21 ± 10.75 μV; b-wave,

80.78 ± 20.58 μV; *n* = 12) were comparable to those of WT mice (a-wave, 20.57 ± 6.69 μV; b-wave, 89.02 ± 22.64 μV; *n* = 10).

In summary, the ERG analysis indicated that our dual-AAV SpRY-ABE8e treatment rescued the retinal function of *rd10* mice.

## Dual-AAV SpRY-ABE8e treatment improved vision-guided behavior in *rd10* mice

We have shown that dual-AAV SpRY-ABE8e treatment rescued retinal function in *rd10* mice. However, whether the improvement at the retinal level eventually translated to the improvement in

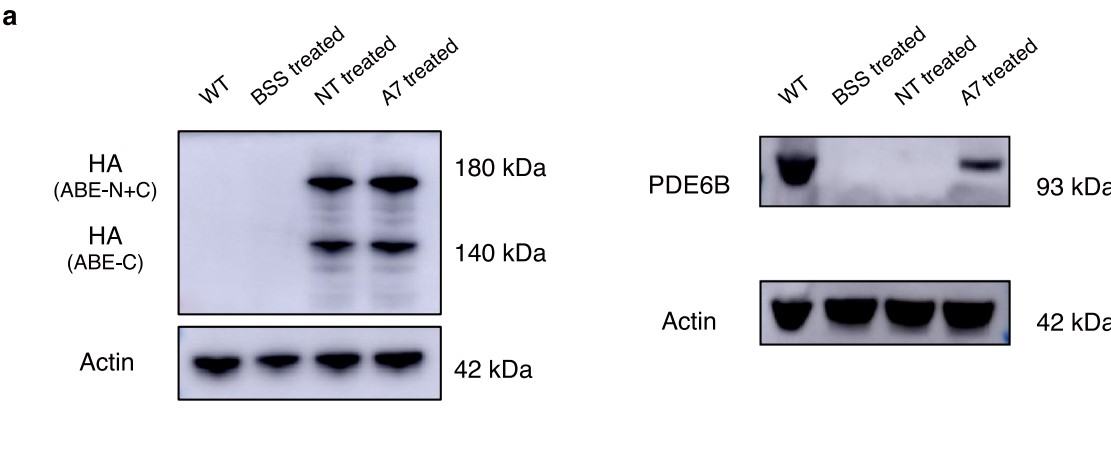

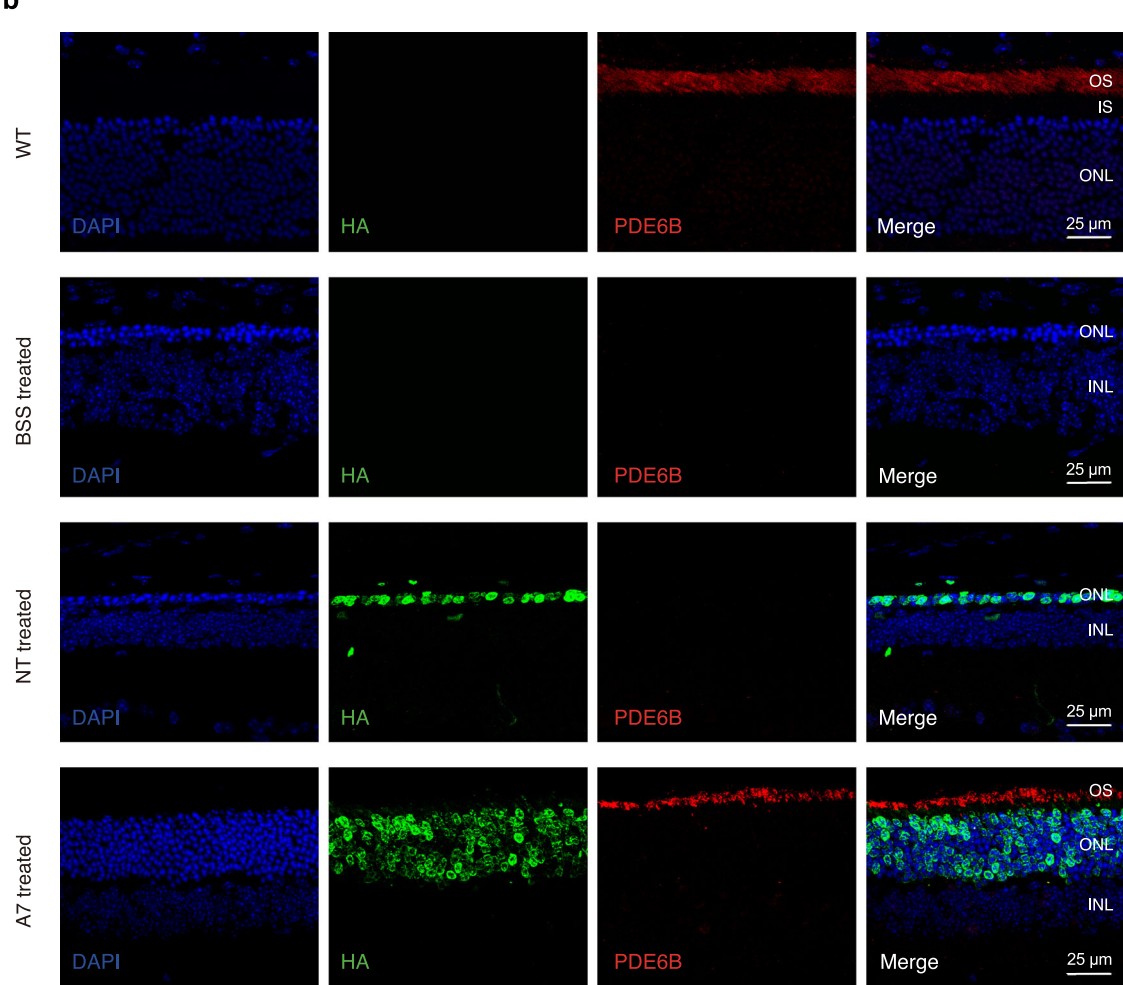

**Fig. 3 | Restoration of PDE6B protein in *rd10* mice after ABE treatment.**
**a** Western blot analysis of ABE (left) and PDE6B (right) expressions in mouse neuroretina lysate after treatment at P35. **b** Immunofluorescence analysis of representative eye cross-sections at P35. Blue indicates DAPI, green indicates HA tag (i.e.,

ABE), and red indicates PDE6B. OS, outer segments; IS, inner segments; ONL, outer nuclear layer; INL, inner nuclear layer. Source data are provided as a Source Data file.

vision-guided behavior remained unknown. To answer this question, we employed the Morris water maze test with modifications (for details please see the Methods section). In this vision-guided behavior assay, mice were tested for their ability to locate a visible platform on 4 consecutive days (D1-D4). The time that the mice took to find the platform was recorded as "escape latency", and the time limit was one minute.

The representative swimming routes on D4 of four testing groups are illustrated in Fig. 6a. During the 4-day period, the success rate of WT mice ($n = 5$) rose up quickly to near 100% (37/40 trials) on D2 (Fig. 6b), and the escape latency gradually decreased to $4.69 \pm 2.13$ s on D4 (Fig. 6c). The control *rd10* mice treated with BSS ($n = 5$) and AAV-SpRY-ABE8e-NT ($n = 5$), however, were rarely able to reach the platform successfully within one minute (12%, 17/140 trials; 9%, 13/140

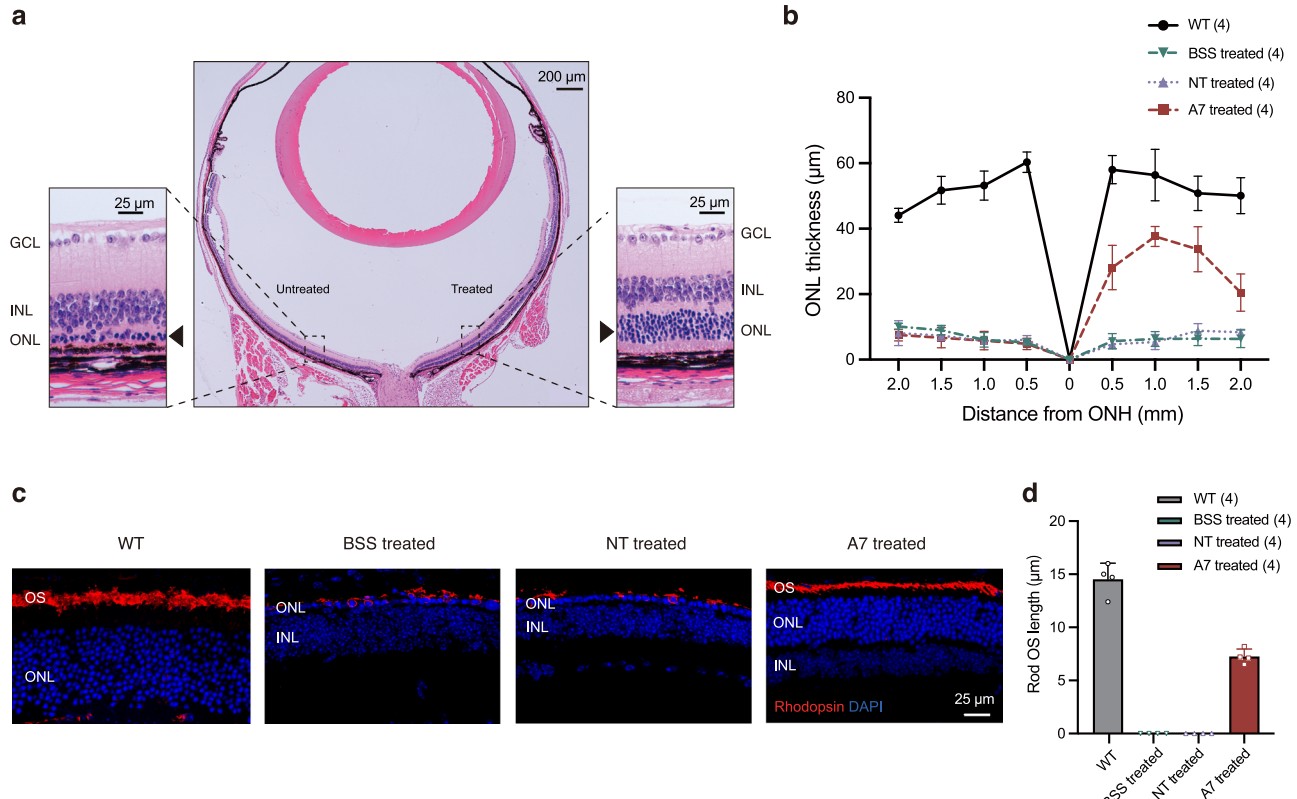

**Fig. 4 | Photoreceptor preservation in *rd10* mice after ABE treatment.**
**a** Representative eye section of an A7-treated *rd10* mouse with H&E staining at P35. ONL, outer nuclear layer; INL, inner nuclear layer; and GCL, ganglion cell layer. **b** Quantification of ONL thickness in DAPI nuclei-stained retinal cryosections of WT (*n* = 4 eyes), BSS treated (*n* = 4 eyes), NT treated (*n* = 4 eyes), and A7 treated (*n* = 4 eyes) mice at P35. ONH, optic nerve head. Data are presented as means ± s.d.

**c** Immunofluorescence analysis of representative retinal sections at P35. Blue indicates DAPI and red indicates Rhodopsin. OS, outer segments. **d** Quantification of rod OS length at 1 mm temporal of the optic nerve head of WT (*n* = 4 eyes), BSS treated (*n* = 4 eyes), NT treated (*n* = 4 eyes), and A7 treated (*n* = 4 eyes) mice at P35. Means ± s.d. are shown. Source data are provided as a Source Data file.

trials, respectively, Fig. 6b) and therefore the escape latency showed no significant decrease over time (Fig. 6c). In contrast to their age-matched counterparts, the AAV-SpRY-ABE8e-A7-treated mice (*n* = 7) showed significantly better performance on the task with the success rate rising steadily from 44% (25/56 trials) on D1 to 85% (48/56 trials) on D3, and all mice successfully located the platform on D4. The corresponding escape latency of AAV-SpRY-ABE8e-A7-treated mice gradually decreased to 18.96 ± 12.99 s by the final testing day (Fig. 6c).

We also analyzed the total path length of the four testing groups and found a tendency similar to the escape latency (Fig. 6d). For WT and AAV-SpRY-ABE8e-A7-treated mice, the path length decreased progressively during the 4-day testing period, while the total path length of BSS-treated and AAV-SpRY-ABE8e-NT-treated mice remained high.

Taken together, these observations indicated that the *rd10* mice were able to process the rescued retinal function properly after our dual-AAV SpRY-ABE8e treatment.

## Discussion

In this work, we developed an AAV-mediated BE therapy for recovering the vision capacity of the *rd10* mouse model. First, a DNA cassette carrying the pathogenic SNV of the *rd10* mouse model was integrated into HEK293T cells, which served as the substrate for testing and optimizing an adenine base editing system. With a dual-AAV system bearing a split SpRY-ABE8e editor employing the intein mechanism, 40.71 ± 1.86% editing efficiency was achieved with minimal bystander editing. *Rd10* mice kept in a dark environment were subretinally injected at P14 with 1 μL of a cocktail of AAV-SpRY-ABE8e at a virus titer of 10^12. At P35, a thick and robust ONL was observed in the AAV-SpRY-

ABE8e-treated area compared with the thin, decayed ONL without treatment. Up to 49% editing/correcting efficiency was obtained in the treated neuroretina. Substantial ERG signals were detected in treated *rd10* eyes, whereas control-treated eyes in the same animals showed minimal signals. Functional rescue was further confirmed by the vision-guided behavior test. During the peer-review process, our findings were supported by an independent investigation[32], which used a different sgRNA design and AAV serotype/dosage, reporting the therapeutic effects of AAV-mediated BE on vision rescue of the *rd10* mouse model, further validating the potential application of base editing on RP treatment.

The development of gene therapies for inherited retinal diseases is of great interest due to the immune-privileged status of the eye and its unique visual and surgical accessibility. Remarkable achievements have been made over the past two decades, highlighted by the first U.S. Food and Drug Administration-approved gene augmentation therapy for RPE65-associated LCA[33]. In recent years, the emergence of CRISPR-based gene-editing tools has offered another promising approach in addition to gene augmentation. CRISPR-Cas nucleases are poorly suited for precisely correcting pathogenic mutations in most therapeutic settings, but CRISPR-based base editors have enabled precise gene correction and disease rescue in multiple preclinical models of genetic disorders. Suh and her colleagues first tested the feasibility in the *rd12* mouse model affected by a nonsense mutation of the *Rpe65* gene (c.130 C > T; p.R44X)[28]. Their landmark work showed that subretinal injection of a lentivirus expressing an ABE and a sgRNA corrected the mutation with up to 29% efficiency. Following treatment, both retinal and visual functions were partially rescued.

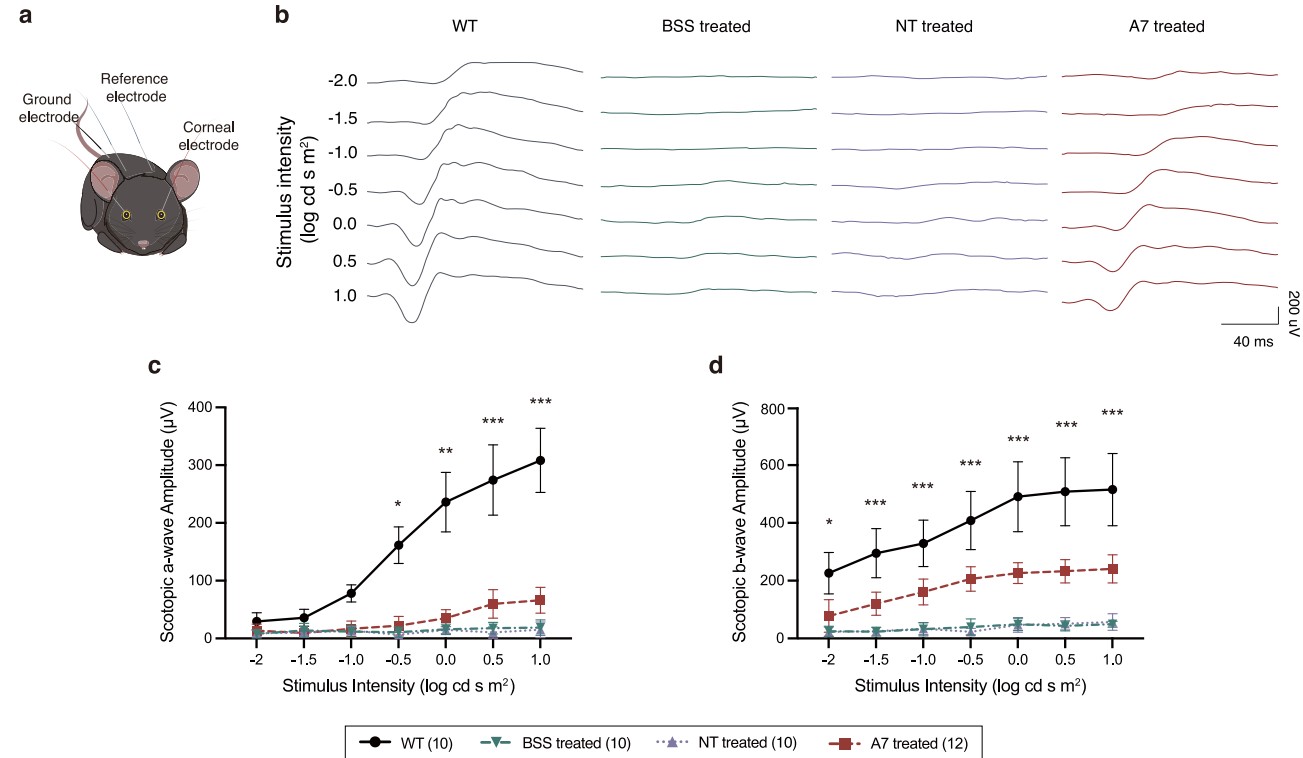

**Fig. 5 | Rescue of retinal function in *rd10* mice after ABE treatment. a** Schematic of in vivo electroretinography settings. **b** Representative scotopic ERG signals of WT, BSS treated, NT treated, and A7 treated eyes at P35. **c** Quantification of scotopic a-wave amplitudes from each group at P35. *$P < 0.05$, **$P < 0.01$, ***$P < 0.001$, two-way ANOVA tests with Tukey's multiple comparisons. Asterisks indicate significant differences between A7 treated and NT treated mice. P values are 0.0292 at −0.5, 0.0024 at 0, and <0.0001 at higher light intensities. **d** Quantification of scotopic b-wave amplitudes from each group at P35. *$P < 0.05$, ***$P < 0.001$, two-way ANOVA tests with Tukey's multiple comparisons. Asterisks indicate significant differences between A7 treated and NT treated mice. *P* values are 0.0294 at −2 and <0.0001 at higher light intensities. The numbers of eyes were as follows: WT, $n = 10$; BSS treated, $n = 10$; NT treated, $n = 10$; and A7 treated, $n = 12$. Means ± s.d. are shown. Source data are provided as a Source Data file.

Our study adds novel information into this topic in terms of two aspects. First, RPE65 is expressed in the retinal pigment epithelium. However, the majority of RP genes are preferentially or even exclusively expressed in rod photoreceptors[5,6]. Our study, therefore, supports the broad clinical utility of BE techniques. Second, the integrative nature of lentivirus[34] makes it a less frequent choice in clinical application. Moreover, lentiviral vectors are poorly effective in photoreceptors[35], limiting their utility for RP treatment. In contrast, the safety and neuro-tropism of AAV-based gene therapies have been substantially validated by basic research and clinical trials. Our dual-AAV system, therefore, provides a more clinically relevant pathway.

Although the improvements in vision are evident in A7-treated *rd10* mice, the durability of BE treatment requires further investigation. Our preliminary data showed that the therapeutic benefit persisted at P90. By 2.5 months post-treatment (P90), the layer of ONL was 3–4 rows in the treated area (Supplementary Fig. 2a), which in sticking contrast to the untreated area in the same eye, where only a discontinuous row of ONL remained (Supplementary Fig. 2a). Functional rescue was confirmed by in vivo ERG (Supplementary Fig. 2b–d) and water maze test (Supplementary Fig. 3). Notably, the treated *rd10* mice were housed within normal light conditions where phototransduction was required during P28 to P90. This indicates that the survival of photoreceptors resulted from the recovery of phototransduction after *Pde6b* correction. Nonetheless, we also observed a decline in therapeutic response. The ONL thickness at P90 (Supplementary Fig. 2a) was approximately 60% of that at P35 (Fig. 4a), and the scotopic ERG b-wave amplitudes reduced to lower than 50% of that at P35 at all light intensities

(Supplementary Fig. 4). The underlying mechanism of the decline is unclear at this time, but possible contributions include persistent expression of BE and/or immune reactions[36,37], as well as innate ultra-sensitive light-dependent photoreceptor cell death in *rd10* mouse, of which the cellular basis has not been fully understood[38,39]. Aiming towards clinical application, an ideal treatment requires both high-magnitude beneficial response and long-term duration. The therapeutic application of BE technology is in its early stage and we hope that our preliminary longitudinal data could inspire future researches on developing more durable approaches.

Apart from durability, further efforts should be made before AAV-mediated BE therapy could be introduced to human patients. First, it is still unclear which serotypes are most effective for BE of photoreceptors, especially in the delivery of dual-AAV vectors, which warrants further investigation. The emerging engineered AAV vectors may provide better choices than AAV5[40]. Additionally, since only one dosage ($1 \times 10^9$ GC/eye) was tested in the current study, detailed dose-response testing is highly needed to characterize the kinetics of in vivo BE activity. The findings will ultimately provide the basis for dose extrapolation to human trials. Moreover, the use of photoreceptor-specific promoters, e.g., RHO, may enhance treatment safety and efficacy by limiting the expression of BE in rods. It is also worth noting that non-viral vectors are emerging as an alternative approach to delivery gene editing agents[41–44]. The non-viral strategies offer the advantage of transient nuclease activity and allow repetitive dosing. Nevertheless, these novel vectors have limited tropism for photoreceptors compared with AAVs at present, requiring further optimization.

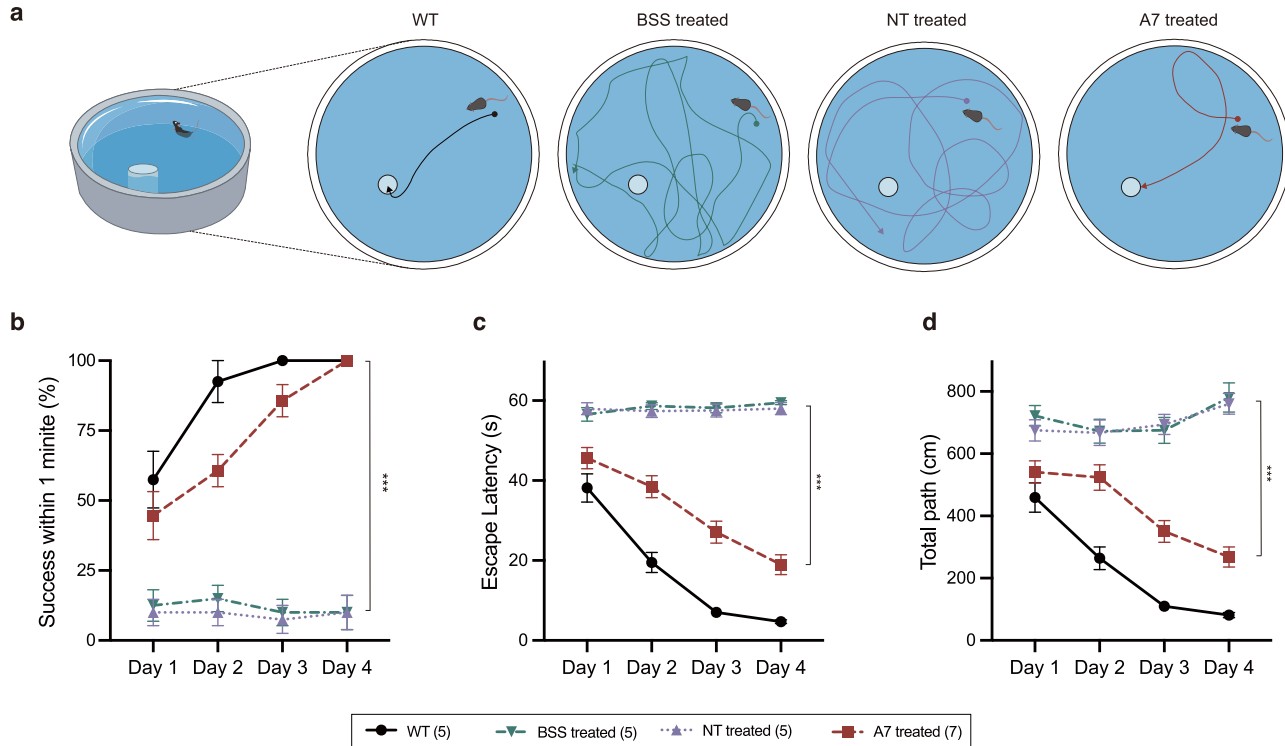

**Fig. 6 | Improvement of vision-guided behavior of *rd10* mice after ABE treatment. a** Representative swimming routes on day 4 from each group at P35. **b** Quantification of the success rate to locate the platform within 1 min from day 1 to day 4. ***$P < 0.001$, two-way ANOVA tests with Tukey's multiple comparisons. Asterisks indicate a significant difference between A7 treated and NT-treated mice at day 4. $P$ value is 0.0004. **c** Quantification of the escape latency from day 1 to day 4. ***$P < 0.001$, two-way ANOVA tests with Tukey's multiple comparisons. Asterisks indicate a significant difference between A7 treated and NT-treated mice at day 4.

$P$ value is <0.0001. **d** Quantification of the total path length from day 1 to day 4. ***$P < 0.001$, two-way ANOVA tests with Tukey's multiple comparisons. Asterisks indicate a significant difference between A7-treated and NT-treated mice at day 4. $P$ value is <0.0001. The numbers of mice were as follows: WT, $n = 5$; BSS treated, $n = 5$; NT treated, $n = 5$; and A7 treated, $n = 7$. Each mouse completed 8 trials on day 1 to day 3 and 4 trials on day 4. Data are shown as the means ± sem. Source data are provided as a Source Data file.

In conclusion, we constructed and demonstrated a feasible therapeutic solution for the treatment of RP. We expect that this work could accelerate the development of BE-based gene therapies.

## Methods

### Ethics statement

This study did not involve human data. All animal procedures were approved by the Institutional Review Board of Shanghai Jiao Tong University and were conducted in accordance with the Association for Research in Vision and Ophthalmology Statement for the Use of Animals in Ophthalmic and Visual Research.

### Animals

Pigmented *rd10* mice and C57BL/6 J mice were obtained from GemPharmatech Co., Ltd and bred at the animal facility of Shanghai General Hospital in 12/12-h light/dark cycles except when interruptions were required for experiments. The temperature ranged from 23–25 °C, and the humidity from 40% to 60%. The mice were housed with 3–5 littermates per cage in individually ventilated cages, with ad libitum access to a normal chow diet and water. The health status of mice were regularly monitored by institutional staff. The sex of mice was not taken into account in the study design. Both female and male mice (approximately 1:1) were used in the experiments. The age, strain, and number of mice used for each experiment are stated in the corresponding figure legends. At the end of the study, mice were euthanized by carbon dioxide exposure.

### Plasmid construction

AAV vectors used for cell transfection or AAV construction were generated with the Gibson Assembly method. Plasmids CBh-V5 AAV-CBE-N (Addgene; #137175) and CBh-V5 AAV-CBE-C (Addgene; #137176) were used to construct different rAAV plasmids. An altered version of the Chicken β-actin (CBA) promoter, CBh (CBA promoter with cytomegalovirus enhancer and hybrid intron)[45], was used to drive the expression of split SpRY-ABE8e in rAAV vectors. The backbones of TadA*-Cas9 (1–573) and Cas9C (574–1368) were subcloned from plasmid SpRY-ABE8e, and the plasmids AAV-SpRY-ABE8e-N and AAV-SpRY-ABE8e-C were constructed by replacing the CBE sequence in the plasmids CBh-V5 AAV-CBE-N and CBh-V5 AAV-CBE-C. The HA tag was fused and expressed at the C-terminus in the AAV-SpRY-ABE8e-C-terminal plasmid. The Pde6b-rd10 c.1678C>T targeted gRNA was installed into the BsmBI sites of the AAV-SpRY-ABE8e-C-terminal plasmid via the Golden Gate method, with the protospacer sequence embedded in the primers. The sgRNA expression was under the control of a U6 promoter. All plasmid constructs were verified by DNA sequencing. AAV vectors were packaged and purified by PackGene Biotech, and the titers were $1 \times 10^{13}$ vg per ml.

### Cell line culture construction and transfection

HEK293T cells (from ATCC) were cultivated in Dulbecco's modified Eagle's medium (DMEM) supplemented with 10% (v/v) fetal bovine serum (FBS) at 37 °C under 5% $CO_2$. A stable cell line containing the mouse Pde6b-rd10 variant was generated as previously described[46]. First, the target sequences were synthesized, amplified, and cloned into the lentiviral vector lentiGuide-Puro (Addgene cat. no. 52963)

using the Golden Gate assembly method. Then, the lentivirus was produced in HEK293T cells using a three-plasmid cotransfection system, including lentiGuide-Puro carrying the target sequences, psPAX2, and pMD2.G. Finally, HEK293T cells were infected with lentivirus to produce a stable cell line containing the integrated target sequences on the genome.

HEK293T cells containing the integrated target sequences were seeded in 24-well plates (Corning, USA). Approximately 24 h after seeding, cells were transfected at approximately 40% confluency with Lipofectamine 2000 (Life Technologies, Invitrogen, USA) according to the manufacturer's protocols. Then, 450 ng of AAV-SpRY-ABE8e-N plasmid and 450 ng of AAV-SpRY-ABE8e-C plasmid were transfected with 50 μl of DMEM containing 1.8 μl of Lipofectamine 2000. At 120 h after transfection, genomic DNA was extracted from the cells using QuickExtract DNA Extraction Solution (Epicenter, USA), and the target genomic regions (200 bp-300 bp) of interest were amplified by PCR for high-throughput DNA sequencing.

### High-throughput DNA sequencing of genomic DNA and cDNA samples

Next-generation sequencing library preparations were constructed following the manufacturer's protocol (VAHTS Universal DNA Library Prep Kit for Illumina), as described previously[5]. Then, libraries with different indexes were multiplexed and loaded on an Illumina HiSeq instrument according to the manufacturer's instructions (Illumina, San Diego, CA, USA). Sequencing was carried out using a 2 × 150 paired-end configuration; image analysis and base calling were conducted by HiSeq Control Software (HCS) + RTA 2.7 (Illumina) on a HiSeq instrument. For paired-end sequencing results, read 1 and read 2 were merged to generate a complete sequence according to their overlapping regions, and a file in FASTA (fa) format was generated. Data were split according to their barcodes. The merged sequences were aligned to the reference sequence by using BWA (version 0.7.12) software. The examined target sites that were mapped with approximately 100,000 independent reads were selected, and obvious base substitutions were observed only at the targeted base editing sites. The base substitution frequencies were calculated by dividing the base substitution read number by the total read number. The associated primers were listed in Supplementary Tables 4 and 5.

### Dark rearing of *rd10* mice

Previous studies found that the *rd10* mouse was sensitive to light exposure[10,12,16,18]. When reared in 12/12-h cyclic lighting conditions, the *rd10* mouse exhibits photoreceptor degeneration starting at approximately postnatal (P) day 16. The number of rows of ONL decays rapidly from 11–12 rows at P20 to 2–3 rows at P25[10,11]. Most photoreceptor cells are lost by P30[10,11]. Conversely, rearing in darkness can delay the onset of photoreceptor degeneration for at least one week[10–12] (Supplementary Fig. 5). To increase the therapeutic window for sufficient transgene expression, we placed late-term pregnant *rd10* females in a 24-h dark room, and their pups were raised under the same conditions until 28 days old, except for the time necessary to perform the subretinal injection (Fig. 2a).

### Subretinal delivery of AAV carrying the SpRY-ABE8e expression cassette

Subretinal injections were performed at P14 with an ophthalmic surgical microscope (Eder Medical Technology, Shanghai, China). The mice were anesthetized by intraperitoneal injection of pentobarbital sodium (50 mg/kg), and their pupils were dilated with 0.5% tropicamide and 0.5% phenylephrine (Mydrin-P, Santen, Osaka, Japan). An incision was made through the corneal limbus at the nasal side using a 32-gauge needle. One microliter of the compound containing AAV-N and AAV-C at a virus titer of $10^{12}$ or balanced salt solution (BSS) was slowly and manually injected into the temporal subretinal space

through the incision using a 36-gauge blunt-end needle mounted on a NanoFil syringe (World Precision Instruments, Florida, USA). An injection was considered successful if the formation of a bleb was visible, indicating a retinal detachment. After the procedure, injected eyes were treated with 0.3% tobramycin and 0.1% dexamethasone (TobraDex, Alcon, Rijksweg, Belgium) to prevent infection. The eyes with substantial complications from the injection procedure, such as hemorrhages or damage to the lens, were excluded from the study and subsequent analysis.

### Western blot analysis

The mouse neuroretinas were harvested and transferred to a tube containing RIPA buffer with protease inhibitors, homogenized with a motor grinder (VCX130, Sonics), and centrifuged for 10 min at 4 °C. The resulting supernatant was mixed with loading buffer (WB2001, NCM Biotech) and heated at 95 °C for 15 min. Each sample was then separated by SDS-PAGE and transferred onto PVDF membranes (catalog ISEQ00010, Millipore), followed by one hour of blocking in 5% nonfat milk in Tris-buffered saline (TBS) containing Tween-20 (TBS-T) at room temperature. Subsequently, the membranes were incubated with primary antibodies at 4 °C overnight, which included rabbit anti-PDE6β antibody (1:1000; catalog PA1-722; Thermo Fisher), rat anti-HA antibody (1:1000, catalog 11867431001, Roche) and rabbit anti-β-actin antibody (1:1000, catalog 3779, ProSci). After overnight incubation, the membranes were washed with TBS-T three times for 10 min each and then incubated with secondary antibodies for 1 h at room temperature, which included HRP-conjugated AffiniPure goat anti-rat IgG (H + L) antibody (1:5000, catalog SA00001-15, Proteintech) and HRP-conjugated AffiniPure goat anti-rabbit IgG (H + L) antibody (1:5000, catalog SA00001-2, Proteintech). After washing with TBS-T three times for 10 min each, the membranes were scanned using an Amersham Imager 600 (GE Healthcare, Freiburg, Germany).

### Immunocytochemistry

Mouse eyes were enucleated and immersed in 4% paraformaldehyde (PFA) in phosphate-buffered saline (PBS) for 10 min, followed by the removal of the anterior segment (cornea and lens). The eyecups were then fixed in 4% PFA for an additional 60 min, washed in PBS three times for 5 min each, and incubated in 30% sucrose in PBS overnight at 4 °C. After incubation, eyecups were placed in a 1:1 mixture of 30% sucrose in PBS and O.C.T. (catalog 4583, Sakura) for one hour and flash-frozen in O.C.T.

Cryosectioning was performed with a Leica CM3050S Cryostat at 12 μm thickness, and the retinal sections were mounted on adhesion slides (catalog 188105, Citotest). The retinal sections were washed with PBS 3 times for 5 min each and blocked for one hour in blocking buffer (1x PBS containing 5% goat serum and 0.3% Triton X-100) at room temperature. After blocking, retinal sections were incubated with rabbit anti-PDE6β antibody (1:500, catalog PA1-722, Thermo Fisher), rat anti-HA antibody (1:500, catalog 11867431001, Roche), rabbit anti-rhodopsin antibody (1:500, catalog 14825 S, CST), and rabbit anti-cone arrestin antibody (1:500, catalog AB15282, Millipore) at 4 °C overnight. After incubation, retinal sections were washed with PBS 3 times for 5 min each before incubation with secondary antibodies, including goat anti-rat IgG (H + L) antibody (1:1000, catalog A-11006, Thermo Fisher) and goat anti-rabbit IgG (H + L) antibody (1:1000, catalog A-11012, Thermo Fisher). Nuclei were stained with DAPI (1:2000, catalog D1306, Thermo Fisher). The slides were then mounted with Fluoromount-G medium (catalog 0100-01, Southern Biotech) and imaged with the Leica SP8 Lightning System.

### Histology

The eyes were fixed in 4% PFA for 24 h and embedded in paraffin, followed by sectioning at 5 μm through the optic disk. Staining was performed with a commercial hematoxylin and eosin kit (catalog

G1003, Servicebio). Slides were imaged using an Olympus BX53 microscope.

## Electroretinography

Full-field electroretinography (ERG) was performed using a RETIanimal System (Roland Consult, Brandenburg, Germany). Before recording, the mice were dark-adapted overnight. The mice were anesthetized by intraperitoneal injection of pentobarbital sodium (50 mg/kg) under dim red illumination. Pupils were dilated with 0.5% tropicamide and 0.5% phenylephrine (Mydrin-P, Santen, Osaka, Japan), and then 0.4% benoxinate hydrochloride (Benoxil, Santen, Osaka, Japan) was applied for local anesthesia. The electrodes were positioned as follows. A pair of gold ring electrodes were placed on the corneal surface of both eyes to serve as active electrodes. Two needle electrodes were inserted between the base of the ear and the lateral canthus to serve as a reference, and one needle electrode was inserted into the base of the tail to serve as a ground electrode. Both eyes were stimulated simultaneously. For scotopic ERG, the eyes were stimulated with a white flash of increasing light intensity (−2.0, −1.5, −1.0, −0.5, 0, 0.5, 1.0 log cd s m$^{-2}$). Then the mice were light-adapted for 10 min and stimulated with white flash at light intensity of 0.5 log cd s m$^{-2}$ for testing photopic ERG. During the procedure, body temperature was maintained at 37 °C with a heating pad. We defined the first negative wave after the flash as the a-wave and the first positive peak occurring after the a-wave trough as the b-wave.

## Vision-guided behavior testing

The Morris water maze was used as a test for vision-dependent behavior with modifications. Briefly, mice were given up to 60 s to locate a visible platform (10 cm in diameter, 0.5 cm above the water level) in a circular pool (1.2 m in diameter, white plastic) filled with opacified water at 23 ± 2 °C. No intentional cues other than the platform were provided. To test mainly rod-mediated vision, mice were pre-dark adapted for at least 12 h, and the experiment was performed under the dim light condition of 0.32 cd/m$^2$ (red light; determined with LS-100, Minolta, Marunouchi, Japan).

The experiment was performed on four consecutive days. Mice were initially trained for three days to learn the task with two blocks of four trials per day. During each trial, the mouse was placed in the water from one of four equally spaced start locations. The start location was changed pseudo-randomly on each of the trials, whereas the platform was kept in a constant location. One trial ended if the mouse climbed onto the platform or if the mouse did not find the platform after 60 s. In the latter case, the mouse would be gently placed onto the platform. The mouse was left on the platform during the inter-trial interval (15 s). After each block, the mouse was towel-dried and transferred to its home cage under warm air. The inter-block interval was approximately one hour. On the subsequent testing day, one block of four trials was performed with the same rules described above. Each mouse completed 28 trials (7 blocks) in total during the 4-day period.

Behavior data were automatically recorded and analyzed with the EthoVision XT tracking system (Noldus, Wageningen, Netherlands). The main outcomes included the time required to reach the platform and the total path length. The latency of mice who didn't find the platform within the time limit (60 s) was recorded as the maximum for data analysis.

## Statistics and reproducibility

Statistical analysis was performed using GraphPad Prism (9.5.0) and Microsoft Excel (Professional Plus 2013) software. The statistical tests used for each experiment are stated in the corresponding figure legends. The experiments in Fig. 3a, b, and Supplementary Fig. 2a were performed in at least three biologically independent animals and similar results were observed.

## Reporting summary

Further information on research design is available in the Nature Portfolio Reporting Summary linked to this article.

## Data availability

The main data supporting the results of this study are available within the paper and in Supplementary Information. High-throughput sequencing data have been deposited in the NCBI database (accession code PRJNA884754). Source data are provided with this paper.

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

## Acknowledgements

This work was supported by the National Natural Science Foundation of China (32171449 to C.B., 82171076 and U22A20311 to X.S., 82000906 to X.W., 32225031 to X.Z., 32271483 and 32001041 to D.Z., 81903776 to J.L.), Tianjin Synthetic Biotechnology Innovation Capacity Improvement Project grants (TSBICIP-KJGG-017 to C.B.), Science and Technology Commission of Shanghai Municipality (20Z11900400 to X.S.), Shanghai Hospital Development Center (SHDC2020CR2040B and SHDC2020CR5014 to X.S.), Shanghai Collaborative Innovation Center for Translational Medicine (CCTS-202202 to X.S.), and Youth Innovation Promotion Association CAS (2022177 to D.Z.), we sincerely thank the Chinese Organization for Rare Disorders for their support of this research.

## Author contributions

X.S., C.B., and X.Z. conceived and designed the research. D.Z., X.C., and Yu.W. designed, performed, and analyzed the in vitro experiments. Yi.W. and X.W. designed, performed and analyzed the in vivo experiments. X.T., J.L., and S.L. performed off-target analysis. C.B., Yi.W., X.W., and D.Z. wrote the manuscript. All of the authors contributed to editing the manuscript.

## Competing interests

The authors declare no competing interests.
