## [Peer Review File · Nature Communications]

REVIEWER COMMENTS

Reviewer #1 (Remarks to the Author):

In this paper the authors explore adenine base editing of DNA in vivo, using the Rd10 mouse which has a retinal degeneration caused by a single point mutation (c.1678C>T) in the beta-6 phosphodiesterase (b6-PDE) enzyme, which regulated the visual cycle. In the naturally occurring mutant mouse, the lack of b6-PDE leads to progressive loss of rod photoreceptors, followed by secondary cone degeneration.

Here the authors use a dual AAV vector approach to deliver a PAM-less, non-cutting Cas9 enzyme (SpRY) fused to an adenine base editor type 8e (ABE8e) to correct the missense A back to G on the reverse strand. This equates to T>C on the plus strand and thereby corrects the error. The editing rate appears to be good on an in vitro study and one guide (sgRNA7) was selected for the in vivo experiment. due to the size constraints, the SpRY-ABE8e sequence was split over two AAV vectors and linked via a split intein bridge to reconstitute the entire protein in vivo.

The data are very good and show excellent anatomical and functional restoration. They show good restoration of ERG function, which is rod dominant, and functional results with the maze swim tests.

Overall, the experiments are well conducted, and I could see no major flaws in the analysis. However, since the CBh promoter was used (please explain the abbreviation), can the authors explain why there is no HA tag seen in the non-photoreceptor cells in figure 3b A7 treated row?

Reviewer #2 (Remarks to the Author):

This study shows the potential efficacy for AAV-mediated base editing to treat patients with retinitis pigmentosa caused by mutations in PDE6B. The authors did an excellent job with their controls and rigor of the experiments, and testing this editing approach has important implications in the field of ophthalmology. However, there are still gaps missing to make the claim of preclinical efficacy and safety of this approach. These are:

Major Comments:

1. The authors state, “This is the first time a human-relevant SNV carried by a mouse model was successfully treated with the safe dual-AAV base editing system in eyes.” However, the use of AAV delivery of base editors to treat LCA has been published in this same journal by another group: Choi, E.H.; Suh, S.; Foik, A.T.; Leinonen, H.; Newby, G.A.; Gao, X.D.; Banskota, S.; Hoang, T.; Du, S.W.; Dong, Z.; et al. *In vivo* base editing rescues cone photoreceptors in a mouse model of early-onset inherited retinal degeneration. *Nat. Commun.* 2022, 13, 1830.

The authors need to remove these overstated claims from their paper.

There is also a strikingly similar study to this one published as a preprint in Biorxiv, thereby reducing the claims that can be made by these authors: Su, J.; She, K.; Song, L.; Jin, X.; Li, R.; Zhao, Q.; Xiao, J.; Chen, D.; Cheng, H.; Lu, F.; et al. *In vivo* base editing rescues photoreceptors in a mouse model of retinitis pigmentosa. *bioRxiv* 2022 doi: <https://doi.org/10.1101/2022.06.20.496770>

2. Figure 4b requires a spider plot in order to show the nuclei thickness spanning from the optic nerve head. This would take into account the treated region that received the subretinal bleb, as well as the untreated portions of the retina. It will also highlight whether the control injections had any effects at the site of the bleb. As a single bar graph, it is unclear whether the authors chose the treated-only portion to quantify, or whether they took into account the entirety of the retina. It also hides any effects the control injections may have had at the site of their subretinal bleb.

3. As this approach edited the *Pde6b* gene in the preclinical mouse model, the authors should show follow-up their H&E, ERG and water maze testing to determine whether this therapy provides a long-lasting efficacy, or whether some degeneration does continue over time (as seen in patients undergoing gene augmentation clinical trials). This is particularly important as the mice were dark-raised for the first month, and follow-up testing after they are housed within normal light conditions requiring phototransduction and PDE6B function is critical. A single time point is not enough to claim long-term efficacy and safety.

4. The authors state that they chose the AAV5 serotype after testing multiple serotypes for their ability to transduce the photoreceptors, and that it was based on published work in the field. However, the previously published work was from 2008, before many serotypes were discovered and tested in the retina. There are many more efficient serotypes for photoreceptor transduction available now. Can the authors show the data for which serotypes they tested to choose AAV5 as the most beneficial, so that it is clear what they compared to their chosen serotype? They should also discuss the limitations/different serotype options in their discussion section.

5. The authors discuss the ability to preserve cones as important for patients with RP, yet they do not test for cone function in their treated mice and controls. Please perform light-adapted ERG to examine the cone function.

6. The authors only tested one dosage for their AAV delivery for its therapeutic efficacy. Please mention this and its implications in the discussion section.

Minor Comments:

1. The authors state that an injection was considered successful if the formation of a bleb was visible directly after injection. Can they confirm that this was their only criteria for injection success? Does this mean they included all mice in their experimental studies, whether or not the bleb caused a permanent retinal detachment? The authors need to include any other exclusion criteria used for their study.

2. The authors use HEK293 cells and HEK293T cells in their manuscript when describing the cell line for testing their base editors. Please fix to reflect the correct cell line used.

3. Please label the retinal cell layers for each of the groups in Figure 3b.

4. The authors state that the mice were dark raised until post-natal day 28 and then moved to standard light conditions and housing. However, their schematic in their figure shows continual dark raising. Can the authors adjust their figure to reflect the correct housing/lighting conditions?

5. Minor grammatical/spelling errors throughout, please correct.

Reviewer #3 (Remarks to the Author):

In this manuscript, Wu et al. use the rd10 mouse model to test a base editing approach with an optimized dual AAV-system. They demonstrated restoration of PDE6B expression, photoreceptor survival (measured by ONL thickness), and rescue of retinal function (shown by ERG and visually-guided behavior). However, there is limited conceptual advance and mechanistic insight.

Comments:

Lines #39-41: The statement in the abstract “Fluorescence microscopy showed that a thick and robust outer nuclear layer (ONL) was developed in the AAV-SpRY41-ABE8e-treated area compared with the thin, underdeveloped ONL without treatment.” is misleading. The ONL was not “developed” in treated areas, but photoreceptor degeneration halted by treatment. ONL was also not “underdeveloped” in untreated areas, but photoreceptor degeneration continued which leads to a decrease in ONL thickness.

Lines #101-102: The statement “Furthermore, the introduced DNA expression cassettes are (...) diluted by cell proliferation” is not true. There is no cell proliferation in the adult retina.

Figure 3a: Immunoblot shows band in NT treated retina. Please provide at least two additional blots.

Figure 4: A more detailed quantification of ONL thickness and OS length, such as via spider plots, and ERG measurements at different light intensities could be informative.

Please indicate the age of the mice and the n-number in each figure legend.

Figure 5b: It is surprising that BSS-treated and NT-treated rd10 mice seem to not find the platform by chance (escape latency at 60 seconds for almost all of the mice). Please provide the escape latency for all 3 days and for all groups. In addition, it would be important to show the total path analysis.

Figure 5 – Title (“Improvement of spatial navigation”) is misleading.

It would be important to analyze the morphology at a later time point (e.g., at 6 months of age) in order to understand the long-term effects of the treatment.

The discussion is rather a summary of the results than a discussion.

Title and line #65 “retinal pigmentosa” should read “retinitis pigmentosa”

Line #73: The acronym RP was already introduced

Figure 1c, Figure 2b: “ariants” (y-axis) should read “variants”

Figure 3: Please indicate the ONL in each row.

Figure 3 and 4: Scale bars are missing.

How were the light conditions in the water maze experiment measured?

Reviewer #1 (Remarks to the Author):

In this paper the authors explore adenine base editing of DNA in vivo, using the Rd10 mouse which has a retinal degeneration caused by a single point mutation (c.1678C>T) in the beta-6 phosphodiesterase (b6-PDE) enzyme, which regulated the visual cycle. In the naturally occurring mutant mouse, the lack of b6-PDE leads to progressive loss of rod photoreceptors, followed by secondary cone degeneration.

Here the authors use a dual AAV vector approach to deliver a PAM-less, non-cutting Cas9 enzyme (SpRY) fused to an adenine base editor type 8e (ABE8e) to correct the missense A back to G on the reverse strand. This equates to T>C on the plus strand and thereby corrects the error. The editing rate appears to be good on an in vitro study and one guide (sgRNA7) was selected for the in vivo experiment. Due to the size constraints, the SpRY-ABE8e sequence was split over two AAV vectors and linked via a split intein bridge to reconstitute the entire protein in vivo.

The data are very good and show excellent anatomical and functional restoration. They show good restoration of ERG function, which is rod dominant, and functional results with the maze swim tests.

Response:

We sincerely appreciate your thorough and accurate summary of the manuscript and encouraging comment.

Overall, the experiments are well conducted, and I could see no major flaws in the analysis. However, since the CBh promoter was used (please explain the abbreviation), can the authors explain why there is no HA tag seen in the non-photoreceptor cells in figure 3b A7 treated row?

Response:

Thank you for raising this out. We have now explained the abbreviation of CBh in the Methods section (Page 21, Lines 424-425).

“An altered version of Chicken β -actin (CBA) promoter, CBh (CBA promoter with cytomegalovirus enhancer and hybrid intron), was used to drive the expression of split SpRY-ABE8e in rAAV vectors.”

Due to the tropism of AAV5, the transgene expression was restricted to photoreceptor and retinal pigment epithelium (RPE) cells despite a ubiquitous promoter (CBh) being used^{1,2}. Because our main purpose was to target photoreceptors where the disease-causing gene (*Pde6b*) expressing, the RPE layer was not shown in Fig. 3b A7 treated row. Below we provided a full view of the retinal cryosection (Fig. R1).

Fig. R1 The expression of ABE8e-HA in an A7-treated eye at P35. **a** Low-magnification image. **b** High magnification image at the red box position of **a**. Blue indicates DAPI and green indicates HA. RPE, retinal pigment epithelium; ONL, outer nuclear layer; INL, inner nuclear layer.

Reviewer #2 (Remarks to the Author):

This study shows the potential efficacy for AAV-mediated base editing to treat patients with retinitis pigmentosa caused by mutations in PDE6B. The authors did an excellent job with their controls and rigor of the experiments, and testing this editing approach has important implications in the field of ophthalmology. However, there are still gaps missing to make the claim of preclinical efficacy and safety of this approach. These are:

Response:

We sincerely appreciate your positive evaluation of our work and constructive suggestions. Below we addressed and clarified your comments in a point-by-point manner, and revised our manuscript accordingly.

Major Comments:

1. The authors state, “This is the first time a human-relevant SNV carried by a mouse model was successfully treated with the safe dual-AAV base editing system in eyes.” However, the use of AAV delivery of base editors to treat LCA has been published in this same journal by another group: Choi, E.H.; Suh, S.; Foik, A.T.; Leinonen, H.; Newby, G.A.; Gao, X.D.; Banskota, S.; Hoang, T.; Du, S.W.; Dong, Z.; et al. In vivo base editing rescues cone photoreceptors in a mouse model of early-onset inherited retinal degeneration. Nat. Commun. 2022, 13, 1830.

The authors need to remove these overstated claims from their paper.

Response:

Thank you for kindly reminding us. In the recently published study, Choi et al. tested the AAV delivery of base editors targeting retinal pigment epithelium (RPE) cells in *rd12* mice³. The series of studies by Prof. Palczewski and his colleagues have pioneered in demonstrating the potential of base editing for treating inherited retinal diseases³⁻⁶. Actually, we were inspired by their works. We have now removed the claims stating “the first” throughout the manuscript.

Removed sentences:

“This is the first time a human-relevant SNV carried by a mouse model was successfully treated with the safe dual-AAV base editing system in eyes” (previous in Abstract, Page 2, Lines 46-47)

“This is the first report of an attempt to develop a market-relevant gene-editing therapy for the treatment of PDE6B-associated RP disease” (previous in Introduction, Page 4, Lines 125-126)

There is also a strikingly similar study to this one published as a preprint in Biorxiv, thereby reducing the claims that can be made by these authors: Su, J.; She, K.; Song, L.; Jin, X.; Li, R.; Zhao, Q.; Xiao, J.; Chen, D.; Cheng, H.; Lu, F.; et al. In vivo base editing rescues photoreceptors in a mouse model of retinitis pigmentosa. bioRxiv 2022 doi: <https://doi.org/10.1101/2022.06.20.496770>

Response:

We sincerely appreciate your careful searching and reminding us that there was a competing work in the meantime. Preprints offer an opportunity to disseminate scientific findings instantly. However, concerns remain on the quality due to loss of editorial evaluation and peer-review process^{7,8}. That's why we decided not to post our manuscript on a preprint server. We are happy that the two independent studies, Su et al.'s⁹ (published during our revision) and ours, arrived at a similar conclusion that AAV-mediated base editing treatment could ameliorate the disease phenotypes in *rd10* mice, validating our findings. This reminds us of the development history of RPE65 gene therapy, a groundbreaking work based on findings from multiple groups over two decades. No doubt reproducibility is highly needed in translational research.

We have now cited Su and colleagues' work in Discussion section (**Page 18, Lines 344-348**).

“Our findings were supported by an independent investigation⁹, which used a different sgRNA design and AAV serotype/dosage, reporting the therapeutic effects of AAV-mediated BE on vision rescue of the *rd10* mouse model, further validating the potential application of base editing on RP treatment.”

2. Figure 4b requires a spider plot in order to show the nuclei thickness spanning from the optic nerve head. This would take into account the treated region that received the subretinal bleb, as well as the untreated portions of the retina. It will also highlight whether the control injections had any effects at the site of the bleb. As a single bar graph, it is unclear whether the authors chose the treated-only portion to quantify, or whether they took into account the entirety of the retina. It also hides any effects the control injections may have had at the site of their subretinal bleb.

Response:

Thanks for your great suggestion. We have now replaced the bar plot with spider plot and rephrased the results (**Page 12, Lines 214-219**). In control treated eyes, the ONL density in treated region was comparable to that in the untreated region, indicating that control injection had no effect on the site of the subretinal bleb.

Fig. 4b Quantification of ONL thickness in DAPI nuclei-stained retinal cryosections of WT (n = 4 eyes), BSS treated (n = 4 eyes), NT treated (n = 4 eyes), and A7 treated (n = 4 eyes) mice at P35. ONH, optic nerve head. Data are presented as means \pm s.d..

“The quantitative measurement on the retinal sections indicated that the ONL was up to $37.62 \pm 3.06 \mu\text{m}$ at the thickest position in the AAV-SpRY-ABE8e-A7-treated eyes (n = 4). This thickness was $\sim 66\%$ of the WT ($56.89 \pm 7.89 \mu\text{m}$, n = 4) while in striking contrast to the BSS-treated (n = 4) and AAV-SpRY-ABE8e-NT-treated eyes (n = 4) where the ONL thickness dropped to averaging $< 11 \mu\text{m}$ (Fig. 4b).”

3. As this approach edited the *Pde6b* gene in the preclinical mouse model, the authors should show follow-up their H&E, ERG and water maze testing to determine whether this therapy provides a long-lasting efficacy, or whether some degeneration does continue over time (as seen in patients undergoing gene augmentation clinical trials). This is particularly important as the mice were dark-raised for the first month, and follow-up testing after they are housed within normal light conditions requiring phototransduction and *PDE6B* function is critical. A single time point is not enough to claim long-term efficacy and safety.

Response:

Thank you for raising this very important and insightful point. In the ongoing clinical trials of RPE65 gene augmentation, patients often showed a continuation of retinal degeneration a few years after treatment¹⁰⁻¹². The reason for the progressive degeneration is unclear, but possible explanations include unmet physiological demand and/or waning of exogenous RPE65 expression, as well as cellular stress from the products of the mutant allele^{5,10-12}. These potential shortcomings to existing gene augmentation have motivated researchers to develop additional strategies for IRDs. Since base editing corrects the disease-causing mutation in its endogenous locus, it is expected to produce long-lasting therapeutic effect. Actually, this hypothesis has been tested in a recent published study by

Choi and colleagues³. Using the *rd12Gnat1*^{-/-} mouse model, they showed that *in vivo* correction of the *rd12* mutation prolonged the survival of cones up to 6 months, which was not achieved with standard gene augmentation³.

In the current proof-of-concept study, we set the time point for analysis at P35 based on previous observation that the dark-reared *rd10* mice experienced a rapid loss of photoreceptors to 1-2 rows after being exposed to 12/12-h cyclic lighting for one week¹³⁻¹⁶. We fully agree with you that durability is a critical point apart from the initial magnitude of therapeutic response. This is of particular importance when aiming at clinical translation. Unfortunately, our follow-up study was severely hindered by COVID-19 pandemic (Shanghai underwent a strict lockdown last year). During the time period of the peer-review process, we replicated the experiment and closely followed up with the treated *rd10* mice. We found that the benefits following treatment persisted but the magnitude was waning over time.

We have now provided our preliminary data and discussed the possible reasons for the decline. These results also highlight the need for further optimization before clinical translation.

In Pages 19-20, Lines 372-392:

“Although the improvements in vision are evident in A7-treated *rd10* mice, the durability of BE treatment requires further investigation. Our preliminary data showed that the therapeutic benefit persisted at P90. By 2.5 months post-treatment (P90), the layer of ONL was 3-4 rows in the treated area (Supplementary Fig. 2a), which in sticking contrast to the untreated area in the same eye, where only a discontinuous row of ONL remained (Supplementary Fig. 2a). Functional rescue was confirmed by *in vivo* ERG (Supplementary Fig. 2b-d) and water maze test (Supplementary Fig. 3). Notably, the treated *rd10* mice were housed within normal light conditions where phototransduction was required during P28 to P90. This indicates that the survival of photoreceptors resulted from the recovery of phototransduction after *Pde6b* correction. Nonetheless, we also observed a decline in therapeutic response. The ONL thickness at P90 (Supplementary Fig. 2a) was approximately 60% of that at P35 (Fig. 4a), and the scotopic ERG b-wave amplitudes reduced to lower than 50% of that at P35 at all light intensities (Supplementary Fig. 4). The underlying mechanism of the decline is unclear at this time, but possible contributions include persistent expression of BE and/or immune reactions^{5,17}, as well as innate ultra-sensitive light-dependent photoreceptor cell death in *rd10* mouse, of which the cellular basis has not been fully understood^{16,18}. Aiming towards clinical application, an ideal treatment requires both high-magnitude beneficial response and long-term duration. The therapeutic application of BE technology is in its early stage and we hope that our preliminary longitudinal data could inspire future researches on developing more durable approaches.”

Fig. S2 Rescue of retinal structure and visual function in *rd10* mice after ABE treatment at P90. **a** Representative eye section of an AAV-ABE-A7-treated *rd10* mouse at P90 with H&E staining. GCL, ganglion cell layer; INL, inner nuclear layer; ONL, outer nuclear layer. **b** Representative scotopic ERG traces in WT, BSS treated, NT treated, and A7 treated mice. **c, d** Quantification of scotopic a- (c) and b-wave (d) amplitudes in WT (n = 6 eyes), BSS treated (n = 6 eyes), NT treated (n = 6 eyes), and A7 treated (n = 10 eyes) mice at P90. Means \pm s.d. are shown. Two-way ANOVA tests with Tukey's multiple comparisons; *** $p < 0.001$.

Fig. S3 Improvement of vision-guided behavior of *rd10* mice after dual-AAV SpRY-ABE8e treatment at P90. **a** Representative swimming routes on day 4 from each group at P90. **b–d** Quantification of the success rate to locate the platform within 1 min (**b**), escape latency (**c**), and total path length (**d**) from day 1 to day 4. The numbers of mice were as follows: WT, $n = 3$; BSS treated, $n = 3$; NT treated, $n = 3$; and A7 treated, $n = 3$. Each mouse completed 8 trials on day 1 to day 3 and 4 trials on day 4. Data are shown as the means \pm sem. Two-way ANOVA tests with Tukey's multiple comparisons; ** $p < 0.01$, *** $p < 0.001$.

Fig. S4 Quantification of scotopic b-wave amplitudes of A7-treated *rd10* mice at different time points. 5-week group, $n=12$ eyes; 3-month group, $n = 10$ eyes. Means \pm s.d. are shown.

4. The authors state that they chose the AAV5 serotype after testing multiple serotypes for their ability to transduce the photoreceptors, and that it was based on published work in the field. However, the previously published work was from 2008, before many serotypes were discovered and tested in the retina. There are many more efficient serotypes for photoreceptor transduction available now. Can the authors show the data for which serotypes they tested to choose AAV5 as the most beneficial, so that it is clear what they compared to their chosen serotype? They should also discuss the limitations/different serotype options in their discussion section.

Response:

Thank you for this valuable comment. In the past 10 years, with the advances of synthetic biology and our better understanding of AAV, novel vectors are emerging with higher transgene efficiency and specificity^{19,20}. You are right and we apologize for our misleading statement.

In the current study, we tested four available serotypes (AAV2, AAV5, AAV8, and AAVDJ²¹) by measuring GFP expression driven by the CBh promoter. The AAV5 was one of the serotypes that efficiently transduced photoreceptors (Fig. R2). Considering that AAV5 was used in an ongoing clinical trial of PDE6B augmentation (NCT03328130), we selected it for implementing the editors.

Fig. R2 Tropisms of four AAV serotypes in WT retina following subretinal injection. Representative low-(a) and high-magnification (b) images of retinal cross sections. ONL, outer nuclear layer.

We realized that the comparison was not rigorous and didn't necessarily reflect the efficiency when delivering dual-AAV vector. The incorrect statement has now been removed and we discussed different serotype options in Discussion section (Page 20, Lines 394-397).

“...it is still unclear which serotypes are most effective for BE of photoreceptors, especially in the delivery of dual-AAV vectors, which warrants further investigation. The emerging engineered AAV vectors may provide better choice than AAV5²².”

5. The authors discuss the ability to preserve cones as important for patients with RP, yet they do not test for cone function in their treated mice and controls. Please perform light-adapted ERG to examine the cone function.

Response:

Thanks for your constructive suggestion. We performed light-adapted ERG and the results are now provided in Page 15, Lines 277-284.

“...we further recorded photopic ERG at $0.5 \log \text{cd s m}^{-2}$ to assess the cone function. It was again obvious that AAV-SpRY-ABE8e-A7-treated rd10 mice showed significantly higher a- and b-wave amplitudes than BSS or AAV-SpRY-ABE8e-NT treated rd10 mice (Supplementary Fig.1b and c). The averaging amplitudes (a-wave, $20.21 \pm 10.75 \mu\text{V}$; b-wave, $80.78 \pm 20.58 \mu\text{V}$; $n = 12$) were comparable to those of WT mice (a-wave, $20.57 \pm 6.69 \mu\text{V}$; b-wave, $89.02 \pm 22.64 \mu\text{V}$; $n = 10$).”

Fig. S1b Representative photopic ERG waveforms at P35. **c** Averaging photopic a- and b-wave amplitudes in WT ($n=10$), BSS treated ($n=10$), NT treated ($n=10$), and A7 treated ($n=12$) eyes. Means \pm s.d. are shown. One-way ANOVA tests with Tukey's multiple comparisons; * $p < 0.05$, ** $p < 0.01$, *** $p < 0.001$.

6. The authors only tested one dosage for their AAV delivery for its therapeutic efficacy. Please mention this and its implications in the discussion section.

Response:

This is an important point made by the reviewer. Further efforts are needed to determine the optimal dosage. We have now discussed this point in Discussion section. Thank you for your great suggestion.

In Page 20, Lines 397-400:

“...since only one dosage (1×10^9 GC/eye) was tested in the current study, a detailed dose-response testing is highly needed to characterize the kinetics of *in vivo* BE activity. The findings will ultimately provide the basis for dose extrapolation to human trials.”

Minor Comments:

1. *The authors state that an injection was considered successful if the formation of a bleb was visible directly after injection. Can they confirm that this was their only criteria for injection success? Does this mean they included all mice in their experimental studies, whether or not the bleb caused a permanent retinal detachment? The authors need to include any other exclusion criteria used for their study.*

Response:

Thank you for prompting us to clarify this point. We excluded the eyes with substantial complications from the injection procedure, such as hemorrhages or damage to the lens. Since all of the injection was performed by an experienced ophthalmic surgeon, the rate of such cases is very low. We have now clarified the exclusion criteria at **Lines 502-504**.

“The eyes with substantial complications from the injection procedure, such as hemorrhages or damage to the lens, were excluded from the study and subsequent analysis.”

The bleb typical resolved within 3 days^{23,24}. Actually, the mouse retina is easy for reattachment²⁵ and therefore we²⁶ and other groups²⁷⁻²⁹ use sodium hyaluronate to prevent reattachment when establishing retinal detachment mouse model.

2. *The authors use HEK293 cells and HEK293T cells in their manuscript when describing the cell line for testing their base editors. Please fix to reflect the correct cell line used.*

Response:

We used HEK293T cells for *in vitro* testing. Thank you for catching this typo. Now corrected.

3. *Please label the retinal cell layers for each of the groups in Figure 3b.*

Response:

Now added. Thanks.

4. The authors state that the mice were dark raised until post-natal day 28 and then moved to standard light conditions and housing. However, their schematic in their figure shows continual dark raising. Can the authors adjust their figure to reflect the correct housing/lighting conditions?

Response:

Thanks for pointing this out. An updated Fig. 2a was provided now.

Fig. 2a Flowchart of *in vivo* ABE treatment.

5. Minor grammatical/spelling errors throughout, please correct.

Response:

Sorry for these language issues. The manuscript has now been proofread by a native English speaker.

Reviewer #3 (Remarks to the Author):

In this manuscript, Wu et al. use the rd10 mouse model to test a base editing approach with an optimized dual AAV-system. They demonstrated restoration of PDE6B expression, photoreceptor survival (measured by ONL thickness), and rescue of retinal function (shown by ERG and visually-guided behavior). However, there is limited conceptual advance and mechanistic insight.

Response:

Thank you for reviewing our manuscript in detail and your constructive critiques. Below we provided point-by-point responses to your comments and suggestions. All of them are valuable, which have undoubtedly improved the quality of our manuscript.

Comments:

Lines #39-41: The statement in the abstract “Fluorescence microscopy showed that a thick and robust outer nuclear layer (ONL) was developed in the AAV-SpRY41-ABE8e-treated area compared with the thin, underdeveloped ONL without treatment.” is misleading. The ONL was not “developed” in treated areas, but photoreceptor degeneration halted by treatment. ONL was also not “underdeveloped” in untreated areas, but photoreceptor degeneration continued which leads to a decrease in ONL thickness.

Response:

Thank you for identifying our inaccurate description. Now corrected.

In Abstract, Page 3, Lines 38-39:

“Fluorescence microscopy showed that a thick and robust outer nuclear layer (ONL) was preserved in the AAV-SpRY-ABE8e-treated area compared with the thin, degenerated ONL without treatment.”

Lines #101-102: The statement “Furthermore, the introduced DNA expression cassettes are (...) diluted by cell proliferation” is not true. There is no cell proliferation in the adult retina.

Response:

We apologized for the wrong information. Now modified (**Page 5, Lines 97-98**).

“Furthermore, the introduced DNA expression cassettes may degrade over time, which causes the fading effect^{11,12,30}.”

Figure 3a: Immunoblot shows band in NT treated retina. Please provide at least two additional blots.

Response:

Three additional blots are provided below. We did not find PDE6B band in either NT-treated or BSS-treated eyes.

The Figure 3a has now been updated. Thanks for your careful reviewing.

Fig. R3 Western blot analysis to detect PDE6B expression in mouse neuroretina lysate after treatment at P35. Three biologically independent replicates were performed.

Figure 4: A more detailed quantification of ONL thickness and OS length, such as via spider plots, and ERG measurements at different light intensities could be informative.

Response:

Thanks for your constructive suggestions. We have now replaced the bar plot with spider plot. We also immunolabeled retinal sections with rhodopsin antibody to measure OS.

In Page 12, Lines 214-226:

“The quantitative measurement on the retinal sections (Fig. 4b) indicated that the ONL was up to $37.62 \pm 3.06 \mu\text{m}$ at the thickest position in the AAV-SpRY-ABE8e-A7-treated eyes ($n = 4$). This thickness was ~66% of the WT ($56.89 \pm 7.89 \mu\text{m}$, $n = 4$), in contrast to the BSS-treated ($n = 4$) and AAV-SpRY-ABE8e-NT-treated eyes ($n = 4$) where the ONL thickness dropped to averaging $< 11 \mu\text{m}$ (Fig. 4b).

To measure the rod outer segments (OS), we immunolabeled retinal sections with rhodopsin antibody (Fig. 4c). For AAV-SpRY-ABE8e-A7-treated eyes, the OS was clearly visible and the length ($7.25 \pm 0.7 \mu\text{m}$, $n = 4$) was ~50% of the WT ($14.53 \pm 1.5 \mu\text{m}$, $n = 4$) at thickest ONL portion (1 μm temporal of the optic nerve head; Fig. 4b, 4d). In BSS-treated and AAV-SpRY-ABE8e-NT-treated eyes, we could not identify typical OS structure, and we found that rhodopsin redistributed from outer segments to inner segments and photoreceptor cell bodies (Fig. 4c).”

Fig. 4 Photoreceptor preservation in *rd10* mice after ABE treatment. **a** Representative eye section of an A7-treated *rd10* mouse with H&E staining at P35. ONL, outer nuclear layer; INL, inner nuclear layer; and GCL, ganglion cell layer. **b** Quantification of ONL thickness in DAPI nuclei-stained retinal cryosections of WT (n = 4 eyes), BSS treated (n = 4 eyes), NT treated (n = 4 eyes), and A7 treated (n = 4 eyes) mice at P35. ONH, optic nerve head. Data are presented as means ± s.d.. **c** Immunofluorescence analysis of representative retinal sections at P35. Blue indicates DAPI and red indicates Rhodopsin. OS, outer segments. **d** Quantification of rod OS length at 1μm temporal of the optic nerve head of WT (n = 4 eyes), BSS treated (n = 4 eyes), NT treated (n = 4 eyes), and A7 treated (n = 4 eyes) mice at P35. Means ± s.d. are shown.

Following your suggestion, we recorded scotopic ERGs in dark-adapted mice using a series of light stimuli increasing from -2 to 1.0 log cd s m⁻².

In Pages 13-14, Lines 253-267:

“To analyze the rod-dominated retinal function in more detail, we recorded scotopic ERGs in dark-adapted mice using a series of light stimuli increasing from -2 to 1.0 log cd s m⁻². The representative scotopic ERG traces and corresponding quantifications of the a- and b-wave amplitudes were shown in Figure 5. The rod-dominated responses from age-matched WT mice showed an expected steady increase in amplitudes of both a- and b-waves, with the increasing stimulus intensity. As for the control *rd10* mice treated with BSS and AAV-SpRY-ABE8e-NT, the a-wave amplitudes attenuated to a negligible level, and b-wave amplitudes were also strongly reduced throughout the range of light stimuli. In the AAV-SpRY-ABE8e-A7-treated *rd10* mice, however, the b-waves were

clearly visible at all stimulus intensities, and the a-waves were evident when the stimulus intensity rose up to 0 log cd s m⁻². The averaging a- and b-wave amplitudes were 66.11 ± 22.27 and 241.17 ± 48.60 μV for AAV-SpRY-ABE8e-A7-treated eyes (n = 12) as the light intensity achieved 1.0 log cd s m⁻², which were ~21% and 47% of those from age-matched WT eyes (a-wave, 308.30 ± 55.57 μV; b-wave, 515.50 ± 125.14 μV; n = 10).”

Fig. 5 Rescue of retinal function in *rd10* mice after ABE treatment. **a** Schematic of *in vivo* electroretinography settings. **b** Representative scotopic ERG signals of WT, BSS treated, NT treated, and A7 treated eyes at P35. **c, d** Quantification of scotopic a- (c) and b-wave (d) amplitudes of WT (n = 10 eyes), BSS treated (n = 10 eyes), NT treated (n = 10 eyes), and A7 treated (n = 12 eyes) mice at P35. Means ± s.d. are shown. Two-way ANOVA tests with Tukey’s multiple comparisons; *** $p < 0.001$.

Please indicate the age of the mice and the n-number in each figure legend.

Response:

We have now added the information in each figure legend. Thanks for pointing this out.

Figure 5b: It is surprising that BSS-treated and NT-treated *rd10* mice seem to not find the platform by chance (escape latency at 60 seconds for almost all of the mice). Please provide the escape latency for all 3 days and for all groups. In addition, it would be important to show the total path analysis.

Response:

We apologize for our unclear description. The Morris water maze test was originally introduced as a test for hippocampus-dependent spatial navigation³¹ and used later as a test for vision-guided behavior with modifications (Fig. R4)^{13,32,33}.

Figure R4. Setting of the water maze test. A visible platform (0.01 m²) was placed at one quadrant of the circular pool (1.44 m²). The water was made opaque by the addition of nontoxic white dye.

Yes, you are right. The control treated mice might find the platform by chance but the rate was low (29/280 trials during four days). Please see below for detailed results.

We have now rephrased the methodology in detail and elaborated on the training process of the water maze test according to your suggestions. The escape latency and total path length for all groups on each testing day are provided in the updated Fig. 6.

In Methods, Pages 26-27, Lines 581-597:

“The experiment was performed on four consecutive days. Mice were initially trained for three days to learn the task with two blocks of four trials per day. During each trial, the mouse was placed in the water from one of four equally spaced start locations. The start location was changed pseudo-randomly on each of the trials, whereas the platform was kept in a constant location. One trial ended if the mouse climbed onto the platform or if the mouse did not find the platform after 60 s. In the latter case, the mouse would be gently placed onto the platform. The mouse was left on the platform during the inter-trial interval (15 s). After each block, the mouse was towel-dried and transferred to its home cage under warm air. The inter-block interval was approximately one hour. On the subsequent testing day, one block of four trials was performed with the same rules described above. Each mouse completed 28 trials (7 blocks) in total during the 4-day period.

Behavior data were automatically recorded and analyzed with EthoVision XT tracking system (Noldus, Wageningen, Netherlands). The main outcomes included the time required to reach the platform and the total path length. The latency of mice who didn't find the platform within the time limit (60 s) was recorded as the maximum for data analysis.”

In Results, Pages 15-16, Lines 297-316:

“In this vision-guided behavior assay, mice were tested for their ability to locate a visible platform on 4 consecutive days (D1-D4). The time that the mice took to find the platform was recorded as “escape latency”, and the time limit was one minute.

The representative swimming routes on D4 of four testing groups are illustrated in Fig. 6a. During the 4-day period, the success rate of WT mice ($n = 5$) rose up quickly to near 100% (37/40 trials) on D2 (Fig. 6b), and the escape latency gradually decreased to 4.19 ± 2.13 s on D4 (Fig. 6c). The control *rd10* mice treated with BSS ($n = 5$) and AAV-SpRY-ABE8e-NT ($n = 5$), however, were rarely able to reach the platform successfully within one minute (9%, 12/140 trials; 12%, 17/140 trials, respectively, Fig. 6b) and therefore the escape latency showed no significant decrease over time (Fig. 6c). In contrast to their age-matched counterparts, the AAV-SpRY-ABE8e-A7-treated mice ($n = 7$) showed significantly better performance on the task with the success rate rising steadily from 44% (25/56 trials) on D1 to 85% (48/56 trials) on D3, and all mice successfully located the platform on D4. The corresponding escape latency of AAV-SpRY-ABE8e-A7-treated mice gradually decreased to 18.96 ± 12.10 s by the final testing day (Fig. 6c).

We also analyzed the total path length of the four testing groups and found a tendency similar to the escape latency (Fig. 6d). For WT and AAV-SpRY-ABE8e-A7-treated mice, the path length decreased progressively during the 4-day testing period, while the total path length of BSS-treated and AAV-SpRY-ABE8e-NT-treated mice remained high.

Taken together, these observations indicated that the *rd10* mice were able to process the rescued retinal function properly after our dual-AAV SpRY-ABE8e treatment.”

Fig. 6 Improvement of vision-guided behavior of *rd10* mice after ABE treatment. **a** Representative swimming routes on day 4 from each group at P35. **b–d** Quantification of the success rate to locate the platform within 1 min (**b**), the escape latency (**c**), and the total path length (**d**) from day 1 to day 4. The numbers of mice were as follows: WT, n = 5; BSS treated, n = 5; NT treated, n = 5; and A7 treated, n = 7. Each mouse completed 8 trials on day 1 to day 3 and 4 trials on day 4. Data are shown as the means \pm sem. Two-way ANOVA tests with Tukey’s multiple comparisons; *** $p < 0.001$.

Figure 5 – Title (“Improvement of spatial navigation”) is misleading.

Response:

Thanks for pointing this out. We have now changed the title to “Improvement of vision-guided behavior of *rd10* mice after ABE treatment”.

It would be important to analyze the morphology at a later time point (e.g., at 6 months of age) in order to understand the long-term effects of the treatment.

Response:

Thank you for this important question. Indeed, Reviewer #2 raised the same concern as yours. In the current proof-of-concept study, we set the time point for analysis at P35 based on previous observation that the dark-reared *rd10* mice experienced a rapid loss of photoreceptors to 1-2 rows after being exposed to 12/12-h cyclic lighting for one

week¹³⁻¹⁶. We hold the same opinion with you that the long-term therapeutic effect should be explored. This is of particular importance when aiming at clinical translation. Unfortunately, our follow-up study was severely hindered due to the strict lockdown in Shanghai last year. During the time period of peer-review process, we replicated the experiment and closely followed up with the treated *rd10* mice. Since base editing corrects the disease-causing mutation in its endogenous locus, we hypothesized that the therapeutic effect would be long-lasting. We found that the benefits following treatment persisted but the magnitude was waning over time. We have now provided our preliminary data and discussed the possible reasons for the decline. These results also highlight the need for further optimization before clinical translation.

In Pages 19-20, Lines 372-392:

“Although the improvements in vision are evident in A7-treated *rd10* mice, the durability of BE treatment requires further investigation. Our preliminary data showed that the therapeutic benefit persisted at P90. By 2.5 months post-treatment (P90), the layer of ONL was 3-4 rows in the treated area (Supplementary Fig. 2a), which in sticking contrast to the untreated area in the same eye, where only a discontinuous row of ONL remained (Supplementary Fig. 2a). Functional rescue was confirmed by *in vivo* ERG (Supplementary Fig. 2b-d) and water maze test (Supplementary Fig. 3). Notably, the treated *rd10* mice were housed within normal light conditions where phototransduction was required during P28 to P90. This indicates that the survival of photoreceptors resulted from the recovery of phototransduction after *Pde6b* correction. Nonetheless, we also observed a decline in therapeutic response. The ONL thickness at P90 (Supplementary Fig. 2a) was approximately 60% of that at P35 (Fig. 4a), and the scotopic ERG b-wave amplitudes reduced to lower than 50% of that at P35 at all light intensities (Supplementary Fig. 4). The underlying mechanism of the decline is unclear at this time, but possible contributions include persistent expression of BE and/or immune reactions^{5,17}, as well as innate ultra-sensitive light-dependent photoreceptor cell death in *rd10* mouse, of which the cellular basis has not been fully understood^{16,18}. Aiming towards clinical application, an ideal treatment requires both high-magnitude beneficial response and long-term duration. The therapeutic application of BE technology is in its early stage and we hope that our preliminary longitudinal data could inspire future researches on developing more durable approaches.”

Fig. S2 Rescue of retinal structure and visual function in *rd10* mice after ABE treatment at P90. **a** Representative eye section of an AAV-ABE-A7-treated *rd10* mouse at P90 with H&E staining. GCL, ganglion cell layer; INL, inner nuclear layer; ONL, outer nuclear layer. **b** Representative scotopic ERG traces in WT, BSS treated, NT treated, and A7 treated mice. **c, d** Quantification of scotopic a- (c) and b-wave (d) amplitudes in WT (n = 6 eyes), BSS treated (n = 6 eyes), NT treated (n = 6 eyes), and A7 treated (n = 10 eyes) mice at P90. Means \pm s.d. are shown. Two-way ANOVA tests with Tukey's multiple comparisons; *** $p < 0.001$.

Fig. S3 Improvement of vision-guided behavior of *rd10* mice after dual-AAV SpRY-ABE8e treatment at P90. **a** Representative swimming routes on day 4 from each group at P90. **b–d** Quantification of the success rate to locate the platform within 1 min (**b**), escape latency (**c**), and total path length (**d**) from day 1 to day 4. The numbers of mice were as follows: WT, $n = 3$; BSS treated, $n = 3$; NT treated, $n = 3$; and A7 treated, $n = 3$. Each mouse completed 8 trials on day 1 to day 3 and 4 trials on day 4. Data are shown as the means \pm sem. Two-way ANOVA tests with Tukey's multiple comparisons; ** $p < 0.01$, *** $p < 0.001$.

Fig. S4 Quantification of scotopic b-wave amplitudes of A7-treated *rd10* mice at different time points. 5-week group, $n=12$ eyes; 3-month group, $n = 10$ eyes. Means \pm s.d. are shown.

The discussion is rather a summary of the results than a discussion.

Response:

We agree with your perspective. We have now added a broader discussion, including the long-term therapeutic effect (In Pages 19-20, Lines 372-392, Please see above response) and future works needed before clinical translation (Pages 20, Lines 393-406) to the Discussion.

In Discussion, Pages 20, Lines 393-406:

“Apart from durability, further efforts should be made before AAV-mediated BE therapy could be introduced to human patients. First, it is still unclear which serotypes are most effective for BE of photoreceptors, especially in the delivery of dual-AAV vectors, which warrants further investigation. The emerging engineered AAV vectors may provide better choices than AAV5²². Additionally, since only one dosage (1×10^9 GC/eye) was tested in the current study, a detailed dose-response testing is highly needed to characterize the kinetics of *in vivo* BE activity. The findings will ultimately provide the basis for dose extrapolation to human trials. Moreover, the use of photoreceptor-specific promoters, e.g., RHO, may enhance treatment safety and efficacy by limiting the expression of BE in rods. It is also worth noting that non-viral vectors are emerging as an alternative approach to delivery gene editing agents³⁴⁻³⁷. The non-viral strategies offer the advantage of transient nuclease activity and allow repetitive dosing. Nevertheless, these novel vectors have limited tropism for photoreceptors compared with AAVs at present, requiring further optimization.”

Title and line #65 “retinal pigmentosa” should read “retinitis pigmentosa”

Response:

Thank you for catching this typo. Now corrected.

Line #73: The acronym RP was already introduced

Response:

Thanks for your careful reviewing. Now corrected.

Figure 1c, Figure 2b: “ariants” (y-axis) should read “variants”

Response:

Sorry for the typo. Now corrected. Thanks.

Figure3: Please indicate the ONL in each row.

Response:

We have now indicated the ONL in each row according to your suggestion.

Figure 3 and 4: Scale bars are missing.

Response:

Now added. Thanks.

How were the light conditions in the water maze experiment measured?

Response:

The light condition was determined using a luminance meter (LS-100, Minolta, Marunouchi, Japan). This detail has now been added into Methods (Page 26, Lines 579-580).

Reference

- 1 Pang, J. J. *et al.* Comparative analysis of in vivo and in vitro AAV vector transduction in the neonatal mouse retina: effects of serotype and site of administration. *Vision Res.* **48**, 377-385, doi:10.1016/j.visres.2007.08.009 (2008).
- 2 Schön, C., Biel, M. & Michalakis, S. Retinal gene delivery by adeno-associated virus (AAV) vectors: Strategies and applications. *Eur. J. Pharm. Biopharm.* **95**, 343-352, doi:10.1016/j.ejpb.2015.01.009 (2015).
- 3 Choi, E. H. *et al.* In vivo base editing rescues cone photoreceptors in a mouse model of early-onset inherited retinal degeneration. *Nature Communications* **13**, 1830, doi:10.1038/s41467-022-29490-3 (2022).
- 4 Suh, S. *et al.* Restoration of visual function in adult mice with an inherited retinal disease via adenine base editing. *Nat Biomed Eng* **5**, 169-178, doi:10.1038/s41551-020-00632-6 (2021).
- 5 Suh, S., Choi, E. H., Raguram, A., Liu, D. R. & Palczewski, K. Precision genome editing in the eye. *Proc. Natl. Acad. Sci. U. S. A.* **119**, e2210104119, doi:10.1073/pnas.2210104119 (2022).
- 6 Yan, A. L., Du, S. W. & Palczewski, K. Genome editing, a superior therapy for inherited retinal diseases. *Vision Res.* **206**, 108192, doi:10.1016/j.visres.2023.108192 (2023).
- 7 Maslove, D. M. Medical Preprints-A Debate Worth Having. *JAMA* **319**, 443-444, doi:10.1001/jama.2017.17566 (2018).
- 8 Flanagan, A., Fontanarosa, P. B. & Bauchner, H. Preprints Involving Medical Research-Do the Benefits Outweigh the Challenges? *JAMA* **324**, 1840-1843, doi:10.1001/jama.2020.20674 (2020).
- 9 Su, J. *et al.* In vivo base editing rescues photoreceptors in a mouse model of retinitis pigmentosa. *Mol Ther Nucleic Acids* **31**, 596-609, doi:10.1016/j.omtn.2023.02.011 (2023).
- 10 Cideciyan, A. V. *et al.* Human retinal gene therapy for Leber congenital amaurosis shows advancing retinal degeneration despite enduring visual improvement. *Proc. Natl. Acad. Sci. U. S. A.* **110**, E517-525, doi:10.1073/pnas.1218933110 (2013).
- 11 Jacobson, S. G. *et al.* Improvement and decline in vision with gene therapy in childhood blindness. *N. Engl. J. Med.* **372**, 1920-1926, doi:10.1056/NEJMoa1412965 (2015).
- 12 Bainbridge, J. W. *et al.* Long-term effect of gene therapy on Leber's congenital amaurosis. *N. Engl. J. Med.* **372**, 1887-1897, doi:10.1056/NEJMoa1414221 (2015).
- 13 Pang, J. J. *et al.* AAV-mediated gene therapy for retinal degeneration in the rd10 mouse containing a recessive PDEbeta mutation. *Invest. Ophthalmol. Vis. Sci.* **49**, 4278-4283, doi:10.1167/iovs.07-1622 (2008).
- 14 Yao, J. *et al.* Caspase inhibition with XIAP as an adjunct to AAV vector gene-replacement therapy: improving efficacy and prolonging the treatment window. *PLoS One* **7**, e37197, doi:10.1371/journal.pone.0037197 (2012).
- 15 Dong, E., Bachleda, A., Xiong, Y., Osawa, S. & Weiss, E. R. Reduced phosphoCREB in Müller glia during retinal degeneration in rd10 mice. *Mol. Vis.* **23**, 90-102 (2017).
- 16 Weh, E., Scott, K., Wubben, T. J. & Besirli, C. G. Dark-reared rd10 mice experience rapid photoreceptor degeneration with short exposure to room-light during in vivo retinal imaging. *Exp. Eye Res.* **215**, 108913, doi:10.1016/j.exer.2021.108913 (2021).
- 17 Bucher, K., Rodríguez-Bocanegra, E., Daultebekov, D. & Fischer, M. D. Immune responses to retinal gene therapy using adeno-associated viral vectors – Implications for treatment success and safety. *Prog. Retin. Eye Res.* **83**, 100915, doi:<https://doi.org/10.1016/j.preteyeres.2020.100915> (2021).

- 18 Sundar, J. C. *et al.* Rhodopsin signaling mediates light-induced photoreceptor cell death in rd10 mice through a transducin-independent mechanism. *Hum. Mol. Genet.* **29**, 394-406, doi:10.1093/hmg/ddz299 (2020).
- 19 Wang, D., Tai, P. W. L. & Gao, G. Adeno-associated virus vector as a platform for gene therapy delivery. *Nat Rev Drug Discov* **18**, 358-378, doi:10.1038/s41573-019-0012-9 (2019).
- 20 Pupo, A. *et al.* AAV vectors: The Rubik's cube of human gene therapy. *Mol. Ther.* **30**, 3515-3541, doi:10.1016/j.ymthe.2022.09.015 (2022).
- 21 Katada, Y., Kobayashi, K., Tsubota, K. & Kurihara, T. Evaluation of AAV-DJ vector for retinal gene therapy. *PeerJ* **7**, e6317, doi:10.7717/peerj.6317 (2019).
- 22 Peters, C. W., Maguire, C. A. & Hanlon, K. S. Delivering AAV to the Central Nervous and Sensory Systems. *Trends Pharmacol. Sci.* **42**, 461-474, doi:10.1016/j.tips.2021.03.004 (2021).
- 23 Qi, Y. *et al.* Trans-Corneal Subretinal Injection in Mice and Its Effect on the Function and Morphology of the Retina. *PLoS One* **10**, e0136523, doi:10.1371/journal.pone.0136523 (2015).
- 24 Huang, P. *et al.* Subretinal injection in mice to study retinal physiology and disease. *Nat. Protoc.*, doi:10.1038/s41596-022-00689-4 (2022).
- 25 Zeng, R., Zhang, Y., Shi, F. & Kong, F. A novel experimental mouse model of retinal detachment: complete functional and histologic recovery of the retina. *Invest. Ophthalmol. Vis. Sci.* **53**, 1685-1695, doi:10.1167/iovs.11-8241 (2012).
- 26 Guo, Y. *et al.* An improved method for establishment of murine retinal detachment model and its 3D vascular evaluation. *Exp. Eye Res.* **193**, 107949, doi:10.1016/j.exer.2020.107949 (2020).
- 27 Matsumoto, H., Miller, J. W. & Vavvas, D. G. Retinal detachment model in rodents by subretinal injection of sodium hyaluronate. *J Vis Exp*, doi:10.3791/50660 (2013).
- 28 Okunuki, Y. *et al.* Microglia inhibit photoreceptor cell death and regulate immune cell infiltration in response to retinal detachment. *Proc. Natl. Acad. Sci. U. S. A.* **115**, E6264-e6273, doi:10.1073/pnas.1719601115 (2018).
- 29 Choi, J. A., Kim, Y. J., Seo, B. R., Koh, J. Y. & Yoon, Y. H. Potential Role of Zinc Dyshomeostasis in Matrix Metalloproteinase-2 and -9 Activation and Photoreceptor Cell Death in Experimental Retinal Detachment. *Invest. Ophthalmol. Vis. Sci.* **59**, 3058-3068, doi:10.1167/iovs.17-23502 (2018).
- 30 Muhuri, M., Levy, D. I., Schulz, M., McCarty, D. & Gao, G. Durability of transgene expression after rAAV gene therapy. *Mol. Ther.* **30**, 1364-1380, doi:<https://doi.org/10.1016/j.ymthe.2022.03.004> (2022).
- 31 Vorhees, C. V. & Williams, M. T. Morris water maze: procedures for assessing spatial and related forms of learning and memory. *Nat. Protoc.* **1**, 848-858, doi:10.1038/nprot.2006.116 (2006).
- 32 Pang, J. J. *et al.* Gene therapy restores vision-dependent behavior as well as retinal structure and function in a mouse model of RPE65 Leber congenital amaurosis. *Mol. Ther.* **13**, 565-572, doi:10.1016/j.ymthe.2005.09.001 (2006).
- 33 Koch, S. *et al.* Gene therapy restores vision and delays degeneration in the CNGB1(-/-) mouse model of retinitis pigmentosa. *Hum. Mol. Genet.* **21**, 4486-4496, doi:10.1093/hmg/dds290 (2012).
- 34 Raguram, A., Banskota, S. & Liu, D. R. Therapeutic in vivo delivery of gene editing agents. *Cell* **185**, 2806-2827, doi:10.1016/j.cell.2022.03.045 (2022).
- 35 van Haasteren, J., Li, J., Scheideler, O. J., Murthy, N. & Schaffer, D. V. The delivery challenge: fulfilling the promise of therapeutic genome editing. *Nat. Biotechnol.* **38**, 845-855, doi:10.1038/s41587-020-0565-5 (2020).
- 36 Banskota, S. *et al.* Engineered virus-like particles for efficient in vivo delivery of therapeutic proteins. *Cell* **185**, 250-265.e216, doi:10.1016/j.cell.2021.12.021 (2022).

37 Trapani, I., Puppo, A. & Auricchio, A. Vector platforms for gene therapy of inherited retinopathies. *Prog. Retin. Eye Res.* **43**, 108-128, doi:10.1016/j.preteyeres.2014.08.001 (2014).

REVIEWERS' COMMENTS

Reviewer #1 (Remarks to the Author):

The authors have addressed my particular concerns and the new RPE staining image is very helpful in validating the tag and the antibody used.

Reviewer #2 (Remarks to the Author):

The authors have taken steps to respond to each of my prior concerns and comments, and I appreciate their hard work in improving their manuscript. The only minor comment that remains is that a scale bar should be included for the retina sections shown in figures 3 and S1.

Reviewer #3 (Remarks to the Author):

The revised version of this manuscript is improved. The effort the authors have taken to revise the text and add additional experimental data is appreciated.

Minor comments:

Fig. 4b: Please check the scale of x-Axis. May not be correct.

Fig. 4d: "Quantification of rod OS length at 1 μ m temporal of the optic nerve head of [...]"

1 μ m may not be correct.

Reviewer #1 (Remarks to the Author):

The authors have addressed my particular concerns and the new RPE staining image is very helpful in validating the tag and the antibody used.

Response:

We are very pleased that we satisfied your particular concerns.

Reviewer #2 (Remarks to the Author):

The authors have taken steps to respond to each of my prior concerns and comments, and I appreciate their hard work in improving their manuscript. The only minor comment that remains is that a scale bar should be included for the retina sections shown in figures 3 and S1.

Response:

We are very pleased that our revised manuscript satisfied your concerns and showed improvement. We sincerely appreciate your time and valuable suggestions that played a significant role in improving the quality of our study. The scale bars have now been added in Figures 3 and S1.

Reviewer #3 (Remarks to the Author):

The revised version of this manuscript is improved. The effort the authors have taken to revise the text and add additional experimental data is appreciated.

Response:

Thanks for your positive feedback, and we appreciated your constructive suggestions immensely during the previous revision resulting in improvements of our manuscript.

Minor comments:

Fig. 4b: Please check the scale of x-Axis. May not be correct.

Fig. 4d: “Quantification of rod OS length at 1 μ m temporal of the optic nerve head of [...]”

1 μ m may not be correct.

Response:

The unit has now been corrected to ‘mm’. Thanks for your careful reviewing.